# Zinc is a master-regulator of sperm function associated with binding, motility, and metabolic modulation during porcine sperm capacitation

Michal Zigo [1✉], Karl Kerns[1,2], Sidharth Sen [3], Clement Essien [4], Richard Oko[5], Dong Xu [4] & Peter Sutovsky [1,6✉]

Sperm capacitation is a post-testicular maturation step endowing spermatozoa with fertilizing capacity within the female reproductive tract, significant for fertility, reproductive health, and contraception. Recently discovered mammalian sperm zinc signatures and their changes during sperm in vitro capacitation (IVC) warranted a more in-depth study of zinc interacting proteins (further zincoproteins). Here, we identified 1752 zincoproteins, with 102 changing significantly in abundance ($P < 0.05$) after IVC. These are distributed across 8 molecular functions, 16 biological processes, and 22 protein classes representing 130 pathways. Two key, paradigm-shifting observations were made: i) during sperm capacitation, molecular functions of zincoproteins are both upregulated and downregulated within several molecular function categories; and ii) Huntington's and Parkinson's disease pathways were the two most represented, making spermatozoon a candidate model for studying neurodegenerative diseases. These findings highlight the importance of $Zn^{2+}$ homeostasis in reproduction, offering new avenues in semen processing for human-assisted reproductive therapy, identification of somatic-reproductive comorbidities, and livestock breeding.

[1] Division of Animal Sciences, University of Missouri, Columbia, MO 65211, USA. [2] Department of Animal Science, Iowa State University, Ames, IA 50011, USA. [3] Institute for Data Science and Informatics, University of Missouri, Columbia, MO 65201, USA. [4] Department of Electrical Engineering and Computer Science, Bond Life Sciences Center, University of Missouri, Columbia, MO 65211, USA. [5] Department of Biomedical and Molecular Sciences, Queen's University, Kingston, ON K7L 3 N6, Canada. [6] Department of Obstetrics, Gynecology & Women's Health, University of Missouri, Columbia, MO 65211, USA. ✉email: zigom@missouri.edu; SutovskyP@missouri.edu

t is without a doubt that zinc ion ($Zn^{2+}$) has an irreplaceable role in male fertility in both lower and higher-order animals. In the late 50 s, researchers observed that chelation of $Zn^{2+}$ with histidine caused starfish spermatozoa to move vigorously[1,2]. Even though they were not fully aware of the physiological implications at the time, decades of rigorous research have borne fruit and we are now beginning to understand the many roles of zinc in men's health, germ cell sustenance, sperm quality, and fertilization[3]. $Zn^{2+}$ is very abundant throughout the male reproductive tract in humans; ~74.6 in testes, ~92.3 in epididymides, ~679 in the prostate, and 102 in seminal vesicles ($\mu g.g^{-1}$ dry weight, respectively)[4]; and is necessary for almost every step of sperm maturation[5–7]. In spermatozoa, $Zn^{2+}$ can act at different levels, mainly as a part of zinc-containing proteins, such as metalloenzymes, or with zinc-interacting proteins, such as zinc transporters and zinc-regulated channels[5]. Most of the labile $Zn^{2+}$ (as opposed to stably bound zinc that together with labile $Zn^{2+}$ constitute the total cellular zinc) though, appears to be coordinated into the proteins through imidazole groups of histidine and thiols of cysteine[8–10] or plasma membrane lipoproteins[11]. Coordinated zinc can be easily chelated from spermatozoa, which is reflected by zinc signal attenuation, and reloaded by the addition of exogenous $Zn^{2+}$, represented by the subsequent signal restoration[12].

Sperm capacitation, initially reported by two groups of investigators independently[13,14] is a terminal sperm maturation process taking place in the oviductal sperm reservoir[15]. This process is triggered by ovulation, leading to activation of proton channels and $Na^+/HCO_3^-$ transporters resulting in alkalization of spermatozoa and membrane hyperpolarization. The pH-sensitive $Ca^{2+}$ channels such as CatSper[16,17] open which leads to $Ca^{2+}$ influx. Bicarbonate and calcium ions activate soluble adenylyl cyclase (SACY), leading to an increase in protein tyrosine phosphorylation, a previously denoted hallmark of sperm capacitation. This terminal sperm maturation step is accompanied by the removal of decapacitating factors, cholesterol, and zinc[12,18,19]. This results in downstream sperm physiology changes - sperm hyperactivation for detachment from the oviductal epithelium within the sperm reservoir; an increase in membrane fluidity allowing lipid raft reorganization at the apical ridge regions; and acrosomal remodeling in preparation for acrosomal exocytosis, zona pellucida (ZP) penetration, and sperm-oolemma adhesion and fusion[20,21]. In vivo, spermatozoa capacitate asynchronously to maximize the time window of oocyte(s) encountered. Sperm capacitation is a terminal event, either resulting in fertilization or sperm death[22].

The $Zn^{2+}$ ion plays a vital role in sperm capacitation, regulating key events that are responsible for fertilization competency. A high extracellular concentration of $Zn^{2+}$ (2 mM) negatively regulates the flagellar voltage-gated proton channel HVCN1, which regulates CatSper channels that are required for hyperactivation in human[23,24] and possibly in bull[25] spermatozoa. The millimolar concentration of $Zn^{2+}$ also inhibits the conversion of proacrosin to acrosin[26,27] as well as the activities of the following zona lysins: (i) 26S proteasome, (ii) matrix metalloproteinases MMP2 and MMP9[28], and (iii) acrosin[27]. The 26S proteasome has also been reported to participate in capacitation events such as degradation of AKAP3 protein[29]; acrosomal remodeling, protein processing, and compartmentalization[30,31]; and spermadhesin de-aggregation from the sperm surface necessary for sperm detachment from the oviductal epithelium[32]. Further, the 26 S proteasome is physiologically important for the detachment of spermatozoa from oviductal glycan beads[33], which was completely inhibited by 2.5 mM $Zn^{2+}$ concentration[28]. Other mechanisms of $Zn^{2+}$ action on the sperm proteome need to be investigated. A recent study of sperm surface CRISP1 offers a glimpse of how $Zn^{2+}$ binding affects the conformation and, in this case, even oligomerization of proteins involved in the regulation of sperm function during fertilization[34].

The utilization of $Zn^{2+}$ by spermatozoa, essential for their function and sustenance, is likely mediated by a diverse but tightly regulated sub-proteome, composed of proteins that either incorporate $Zn^{2+}$ in their backbone or respond to it. It is thus important to examine the effect of $Zn^{2+}$ on sperm capacitation and fertility to investigate the sperm zinc-dependent and zinc-interacting subproteomes. The objectives of this study were therefore (i) to selectively isolate and identify zinc-binding and interacting proteins from porcine spermatozoa, and with the help of bioinformatics to determine how these may be involved in porcine sperm physiology, and (ii) to identify zincoproteins that change during capacitation and thus may be regulated by $Zn^{2+}$ during this event.

## Results

**Zincoproteins identified in non-capacitated and in vitro capacitated boar spermatozoa.** A fresh, pre-sperm rich fraction of ejaculate ($n = 10$) from a fertile, non-transgenic boar (*S. scrofa*, $n = 1$) was used to conduct the experiment. The pre-sperm rich fraction was split into halves, one half was in vitro capacitated and the other half was used as non-capacitated control.

Using Immobilized Metal Ion Affinity Chromatography (IMAC), zinc interacting/binding proteins extracted from non-capacitated and in vitro capacitated (IVC) porcine spermatozoa were successfully isolated. Mild, non-denaturing detergent was used for protein extraction to preserve zinc-binding motifs. The specificity of zinc interactions was confirmed by chelating $Zn^{2+}$ from both sperm proteins and IMAC column with EDTA before IMAC (Fig. S1). At least five biological replicates of sperm IVC and IMAC were performed, and proteomics was performed in four biological replicates including one pilot and three working replicates that were used for statistical and bioinformatic analyses. Altogether, we have identified 3341 UniProtKB annotated accessions (Supplementary Data 1), which translates into 1752 gene-encoded proteins (1740 VGNC annotated gene products). To confirm the identified proteins were zinc interacting/binding, we used machine learning prediction of zinc-binding amino acid residues to determine potential zinc-binding sites in the identified zincoproteins (Fig. 1 and Supplementary Data 2). The highest frequency of identified zincoproteins was found to have between 6 and 20 amino acid (AA) residues that could bind $Zn^{2+}$.

The identified zincoprotein dataset was assessed by multiple PANTHER analyses[35] (Supplementary Data 3) including molecular function, biological process, cellular component, protein class, and pathway to classify the identified proteins. PANTHER molecular function analysis (Figs. 2a, S2, and Supplementary Data 4) identified 8 different categories with the first three most represented categories being catalytic activity (542 protein hits), binding activity (436 protein hits), and molecular function regulator (67 protein hits). PANTHER biological process analysis (Figs. 2b, S3, and Supplementary Data 4) identified 16 different categories with the first three most represented categories being cellular process (802 protein hits), metabolic process (507 protein hits), and biological regulation (250 protein hits). PANTHER cellular component analysis (Supplementary Data 3) identified 3 different cellular categories: anatomical entity (943 protein hits), intracellular (882 protein hits), and protein-containing complex (303 protein hits). PANTHER protein class analysis (Supplementary Data 3) identified 22 different categories with the first three most represented categories being metabolite interconversion enzyme (396 protein hits), protein modifying enzyme (155 protein hits), and cytoskeletal protein (99 protein hits). Finally,

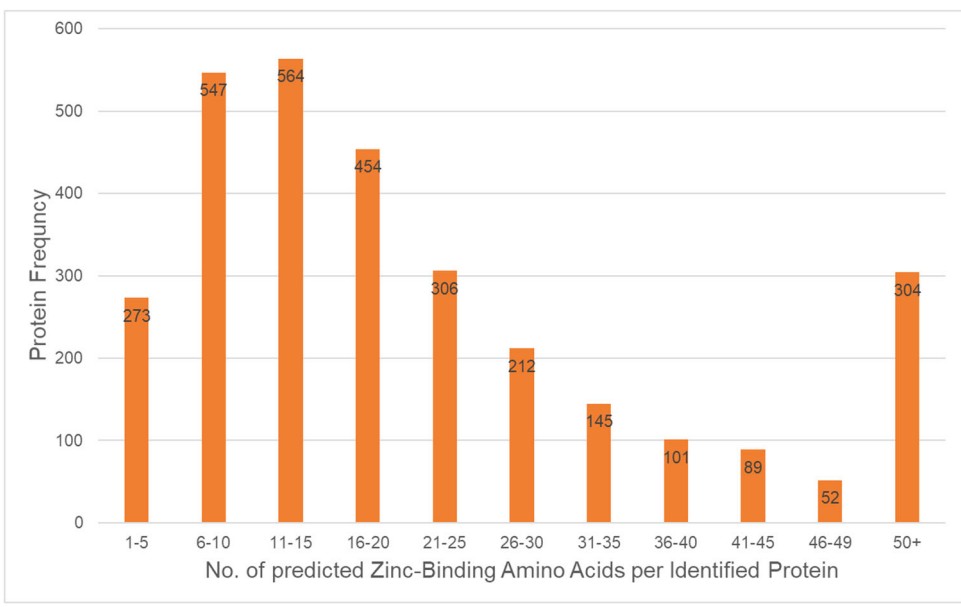

**Fig. 1 Machine learning prediction of zinc-binding AA residues' frequency of pulled-down zincoproteins.** The X-axis represents the number of predicted zinc-binding AA residues including cysteine, histidine, glutamate, and aspartate (CHEDs) within an identified zincoprotein, and the Y-axis represents the number (frequency) of identified zincoproteins within the specified interval of CHEDs.

PANTHER pathway analysis (Supplementary Data 2) identified 130 pathways with the first three most represented categories being Huntington's disease (42 protein hits), Parkinson's disease (33 protein hits), and Wnt signaling pathway (27 protein hits).

Changes in zincoprotein Zn-binding ability during capacitation were analyzed using volcano plot analysis[36], and are presented in Fig. 3; a more detailed plot is presented in Fig. S4. Out of 1752 gene products, the volcano plot analysis found 54 zincoproteins to become significantly less abundant, while 48 zincoproteins became significantly more abundant in spermatozoa subjected to 4 h of IVC when compared to non-capacitated spermatozoa (fold change threshold 2.0; unadjusted $P$-value ≤ 0.05). Both significantly increased and decreased zincoproteins after IVC were subjected to PANTHER molecular function and biological process analyses (Fig. 4 and Supplementary Data 4). PANTHER molecular function analysis (Fig. 4a, c) revealed that within the binding, catalytic activity, and molecular function regulator categories, some functions/activities were being upregulated while others were being downregulated, on top of additional down-regulated functions/activities in IVC spermatozoa. PANTHER biological process analysis (Fig. 4b, d) revealed a similar trend, wherein 7 categories, certain biological processes were upregulated while others were downregulated during sperm capacitation.

**Capacitation related changes of Zincoproteins**. This section is focused on zincoproteins that were found significantly different in abundance levels between IVC and non-capacitated spermatozoa by two-sample, two-tailed $t$-test ($\alpha = 0.05$), making them the prime candidates for processing and regulation during sperm capacitation by $Zn^{2+}$. The list of all 101 identified zincoproteins, including 99 VGNC annotated, and 2 uncharacterized zinco-proteins significantly different in abundance ($P < 0.05$) between IVC and non-capacitated spermatozoa, is presented in Supplementary Data 5. The same PANTHER analyses were performed for capacitation-related, significantly different zincoproteins that are summarized in Supplementary Data 3. A literature search of significantly different zincoproteins was performed by searching each protein individually in the PubMed.gov database for their reported/known function and localization specifically in

spermatozoa (Fig. 5 and Supplementary Data 5). The three most abundant functions in spermatozoa of these proteins were sperm motility (21%), energy metabolism (15%), and cell signaling (13%). Other functions that the significantly different zincopro-teins fulfill in spermatozoa are fertilization (7%); cytoskeleton (5%); pH, redox, or energy homeostasis (4%); proteasome-mediated protein degradation (4%); regulation of immune response, and mitochondrial function (3% both) and nuclear pore transport and $H_2S$ synthesis (both 2%). The remaining zinco-protein functions (4%) include spermatogenesis, steroid bio-synthesis, co-chaperoning, and repair of age-damaged proteins (Fig. 5a). Among significantly different zincoproteins, 17% do not have known function in spermatozoa (no reference found). As for the known localization of significantly different zincoproteins in spermatozoa; these were reported to localize in the whole spermatozoon (13%), or with the exclusive localization in the sperm head (17%), flagellum (19%), or the sperm mitochondria/mid-piece (20%, Fig. 5b). 31% of significantly different zincoproteins have unknown localization in the spermatozoon (no literature reference found). Three significantly different zincoproteins (CA2, CCIN, CCDC39) were phenotyped in non-capacitated and capacitated spermatozoa.

**Localization and dynamics of selected significantly different zincoproteins during sperm capacitation**. Three zincoproteins, namely carbonic anhydrase 2 (CA2), calicin (CCIN), and coiled-coil domain containing 39 (CCDC39), found in significantly different quantities between non-capacitated and IVC sperma-tozoa, were subjected to phenotype studies by the means of image-based flow cytometry (IBFC), immunocytochemistry, and Western blotting (WB) (Figs. 6 and S5–14). Starting with CA2, we found it to be localized across the whole spermatozoa in both non-capacitated and capacitated sperm populations, with the highest fluorescence intensity in the acrosomal and midpiece regions (Figs. 6a and S5a, b). We observed a decrease in the median fluorescence intensity by 40.81 ± 11.85 % after IVC ($P = 0.004$, $n = 5$, paired sample $t$-test, Figs. 6a and S6), as well as a drop in CA2 WB band density by 74.88 ± 23.24 % ($P = 0.01$, $n = 8$, two-sample $t$-test, Figs. 6b and S7) in affinity isolated

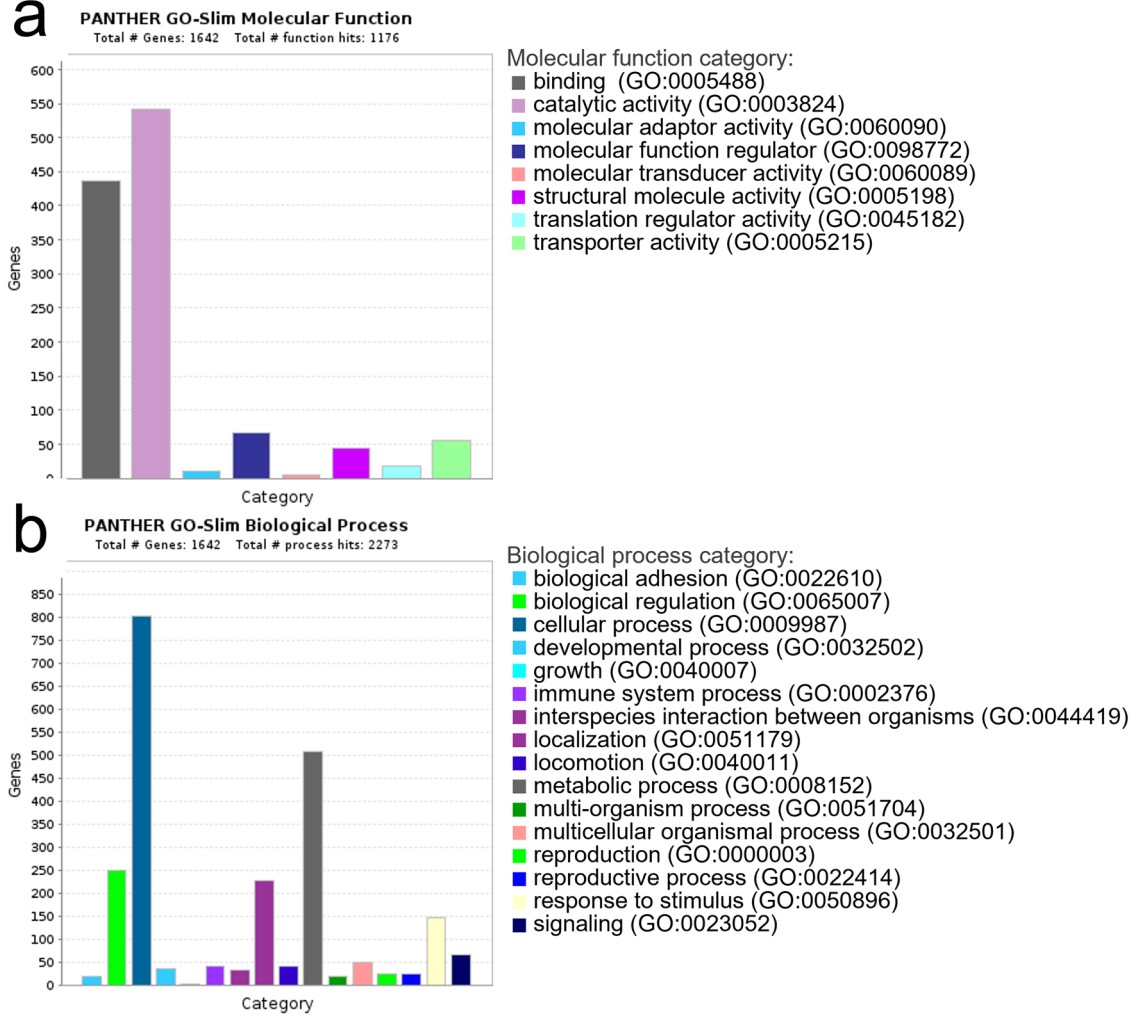

**Fig. 2 Graphic PANTHER analysis view of identified zincoproteins. a** Molecular function and **b** biological process are shown. For clarity, only high-level (level 0) categories are displayed. More detailed graphical presentations, including sub-level (level 1) are shown in Figs. S2 and S3, as well as in Supplementary Data 3.

capacitated sperm zincoproteome when compared to non-capacitated zincoproteome. We observed the same trend in the whole non-capacitated versus capacitated sperm extracts and tracked the origin of CA2 to the testis (Fig. 6c).

The second studied protein, CCIN localized to the post acrosomal sheath (PAS) in non-capacitated spermatozoa, while in capacitated spermatozoa, it localized throughout the entire sperm head region (Figs. 6d and S5c, d). Using IBFC, we detected posterior-to-anterior spread of immunofluorescence signal that was represented by the presence of different CCIN populations in capacitated spermatozoa (Figs. 6d and S8). Since we used rabbit anti-CCIN immune serum prone to non-specific binding in the acrosomal region, we applied a custom image mask (DAPI M07 mask minus PNA-AF647 M11 mask) that excluded the acrosomal area from the calculations to analyze the post-acrosomal segment (PAS) exclusively. We observed a $5.01 \pm 3.13$ fold increase in the median fluorescence intensity in the PAS ($P = 0.016$, $n = 5$, paired sample t-test, Figs. S9a and S10) in IVC spermatozoa. Furthermore, the median fluorescence intensity of the PAS region increased across CCIN subpopulations (Fig. S9b). Similarly, as in CA2, CCIN WB detection was performed within non-capacitated and capacitated zincoproteomes (Figs. 6e and S11). There was a $27.38 \pm 107.91$ % increase in the abundance of CCIN in IVC sperm zincoproteome ($P = 0.513$, $n = 8$, two-sample t-test). Due to

the poor solubility of perinuclear theca proteins, three different additional extraction approaches were performed to increase CCIN extractability, (i) overnight incubation with 1X LDS buffer, (ii) overnight incubation with 0.1 M NaOH, and (iii) sequential extraction as described in[37]. Regardless of the approach, there was a $76.67 \pm 25.94$ % increase in CCIN abundance in IVC sperm extracts ($P = 0.007$, $n = 3$, two-sample t-test, Figs. 6f and S12).

The last studied protein, CCDC39 localized to the flagellum as well as the apical region of the acrosome in both non-capacitated and IVC spermatozoa (Figs. 6g and S5e, f). IBFC determined that the median fluorescence intensity in the apical acrosome area decreased in IVC spermatozoa by $19.07 \pm 8.26$ % ($P = 0.019$, $n = 5$, paired sample t-test, Figs. 6g and S13). Same as with the aforementioned zincoproteins, WB detection of CCDC39 was performed in non-capacitated and capacitated zincoproteomes (Figs. 6h and S14). CCDC39 migrated at ~110 kDa; however, it was beyond the limit of the detection in affinity-purified zincoproteome. Further, CCDC39 was detected in the whole sperm extracts of both non-capacitated and capacitated spermatozoa and was traced to have a testicular origin (Fig. 6i).

## Discussion

Since our recent discovery of capacitation-related sperm zinc ion fluxes[12], our research efforts have been dedicated to furthering

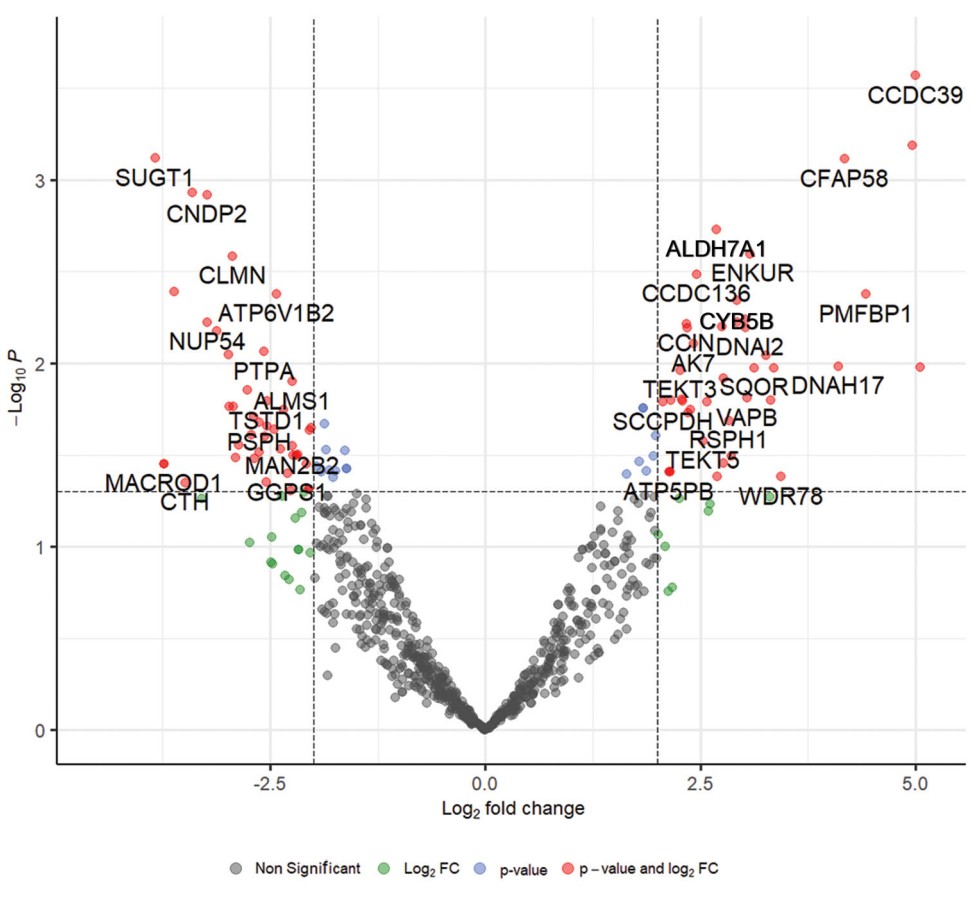

**Fig. 3 Volcano plot representation of changes in zincoprotein abundances before and after 4 h of in vitro capacitation.** Fold change threshold = 2.0 (x-axis) and unadjusted P-value ≤ 0.05 (y-axis). Red circles represent zincoproteins above the fold change and unadjusted P-value threshold; blue circles represent those above the unadjusted P-value threshold but below the fold change threshold; green circles represent zincoproteins above the fold change but below the unadjusted P-value threshold; and gray circles represent those that are below both thresholds.

the understanding of this previously unknown biological phenomenon. In the present study, we set a goal to identify the proteins responsible for zinc-binding and interaction in mammalian spermatozoa and how these change during in vitro capacitation. With the help of IMAC with immobilized $Zn^{2+}$, we successfully isolated zinc-binding /-interacting proteins (zincoproteins) from non-capacitated and IVC spermatozoa, in a highly consistent and replicable manner. Further increasing our confidence in this technique, zinc chelation before IMAC showed that the presence of zinc is essential for the successful zincoproteome pulldown (Fig. S1). The identity of the pulled-down zincoproteins was revealed by high-resolution mass spectrometry. After comparing the identified peptides with UniprotKB *Sus scrofa* protein database entries, we were able to identify 1740 VGNC annotated proteins. We performed machine learning zinc-binding site prediction that gave us high confidence in the zinc-binding/interacting property of identified proteins.

Using PANTHER analyses[35], we explored what protein classes the identified zincoproteins belong to, what molecular functions they fulfill, as well as which biological processes and pathways they are involved in. The three most represented protein classes of the identified zincoproteins were (i) metabolite interconvention enzyme, (ii) protein modifying enzyme, and (iii) cytoskeletal protein. This result is not surprising since $Zn^{2+}$ is the most common cofactor in metalloenzymes[38] and can modulate the activity of other enzymes[28]. This translates into the observation that the most represented molecular function category of the

identified zincoproteins is the catalytic activity, and with the second most represented one, binding, they cumulatively make up for 74% of the identified zincoproteins. We previously proposed that $Zn^{2+}$ from the sperm-induced, oocyte-issued zinc release at fertilization (also known as zinc sparks[39–41]) might act as a fast block to polyspermy[5,12]. Such rapid exocytosis of over 10 billion zinc ions[41] completely inhibits sperm binding to the ZP while sustaining sperm binding to oviductal glycans mimicking oviductal epithelium in an in vitro system[28]. Evolutionary conservation of this so-called oocyte zinc spark[42] corroborates our view that $Zn^{2+}$ can reversibly alter sperm motility and the ability to interact with the oviductal epithelial cells and the egg coat. Cytoskeletal proteins such as actin and tubulin were shown to be zinc-scavenging proteins, and $Zn^{2+}$ stimulates the assembly of tubulin protofilaments[43] and induces F-actin polymerization[44]. Pertinent to spermatozoa, $Zn^{2+}$ binds to nascent outer dense fiber proteins of the sperm flagellum[10] and its flagellar $Zn^{2+}$ concentration was negatively correlated with motility[45]. The identified zincoproteins participate in 16 different biological process categories within the spermatozoa with the three most prominent ones (cumulatively 70%) being biological regulation, cellular process, and metabolic process. Interestingly, out of 130 pathways identified, the two most represented ones were Huntington's and Parkinson's disease. A link between these two diseases and male infertility was established earlier[46,47] offering a new, exquisite model for studying these diseases. Furthermore, testicular function and sperm quality, including motility, is

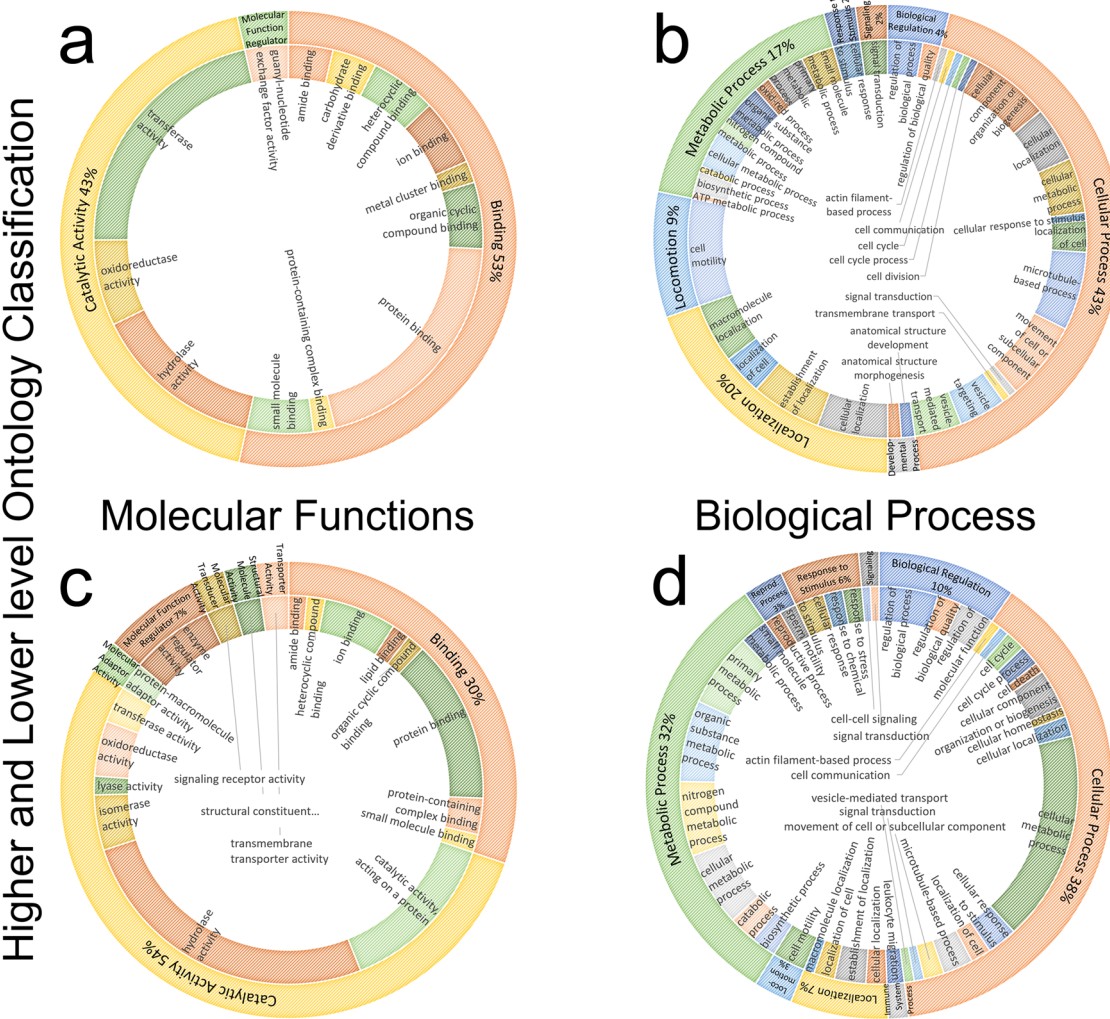

**Fig. 4 PANTHER molecular function and biological process analyses of significantly different zincoproteins after capacitation.** Molecular functions (**a**, **c**) and biological processes (**b**, **d**), differentiated by their higher abundance (**a**, **b**) or lower abundance (**c**, **d**) in zincoproteins found significantly different by the volcano plot analysis. The outer circle represents high-level ontology classification (level 0), while the sub-family (level 1, finer classification of categories) is represented by the inner circle of the respective doughnut chart.

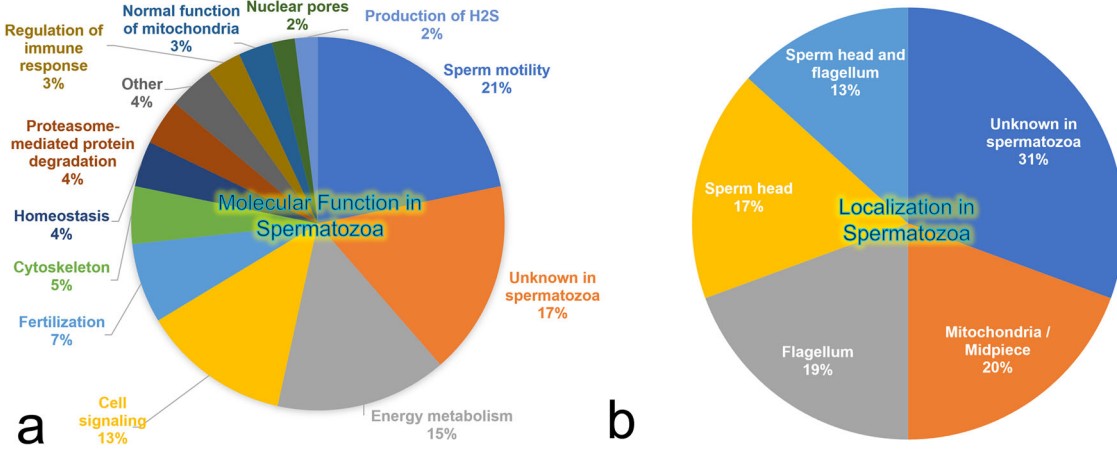

**Fig. 5 Molecular function and localization analysis of significantly different sperm zincoproteins in boar spermatozoa. a** Protein function and **b** cellular localization.

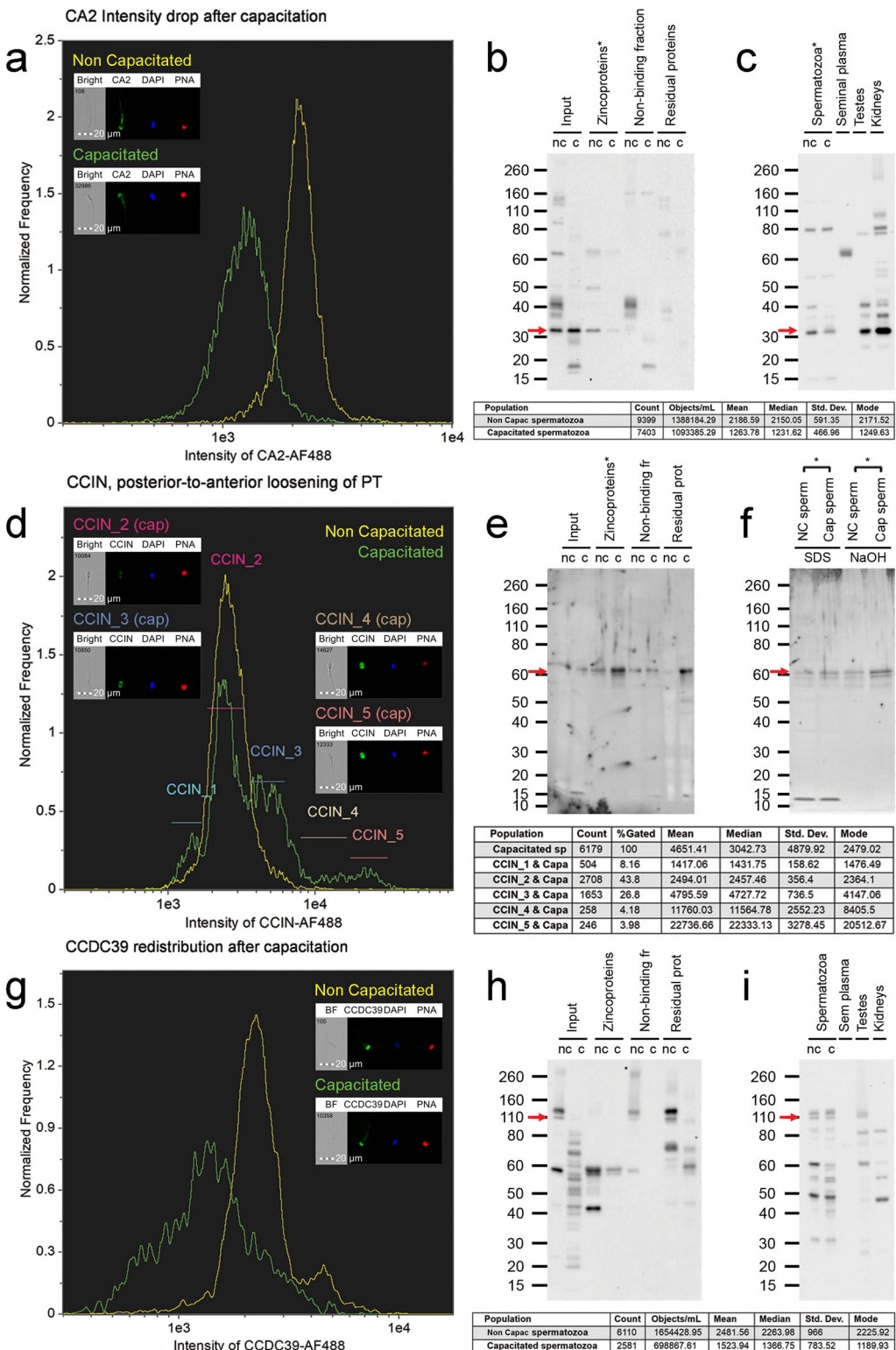

**Fig. 6 Phenotyping of three significantly different zincoproteins between non-capacitated and IVC spermatozoa. a–c** CA2, **d–e** CCIN, and **g–i** CCDC39. IBFC of formaldehyde fixed and Triton X-100 permeabilized (**a**, **d**) or methanol fixed/permeabilized (**g**) spermatozoa; labeled with corresponding primary antibodies. Appropriate species-specific secondary antibodies conjugated to AF488 were used and coincubated with DAPI nuclear stain and PNA-AF647 for acrosome detection. Negative controls with respective non-immune sera of matching immunoglobulin concentration were performed and were published in our previous study. IBFC was performed in 5 replicates with consistent results. WB detection in individual fractions of zincoproteome purification (**b**, **e**, **h**), and WB detection in whole sperm extracts (**c**, **f**, **i**) were employed to fully characterize the capacitation changes in the studied proteins. The red arrows point to the bands of actual (CA2 ~ 32 kDa, CCIN ~ 60 kDa), and predicted (CCDC39 ~ 110 kDa) molecular weights. WB detection in sperm zincoproteome (**b**, **e**, **h**) and the whole sperm extracts (**c**, **f**, **i**) was performed in 8 replicates, and 3 replicates respectively. A statistically significant difference (P < 0.05, two-sample t-test) between non-capacitated and capacitated spermatozoa is indicated by an asterisk.

affected in transgenic porcine and primate models of Huntington's disease[48,49]. This should be further explored with regard to the identification of previously unknown somatic and reproductive system comorbidities, wherein sperm quality and function could serve as a marker of general health and a diagnostic tool for adult-onset diseases, including neurodegeneration. During the peer review of this paper, a study by Zhang et al was published[50] where the authors showed Huntington's and Parkinson's disease pathways being preferentially associated with the perinuclear theca subproteome. The third most represented pathway in our study was Wnt signaling that is intimately involved in male reproductive physiology[51].

To assess the capacitation-related changes in zincoprotein abundance, we performed volcano plot analysis[36] and found the abundance levels of 102 zincoprotein to be significantly different in IVC spermatozoa when compared to non-capacitated control. At this point, it is important to identify the causes of these changes in protein abundance levels. There are at least three: (i) the real change of a protein abundance level without change in zinc affinity, (ii) $Zn^{2+}$ affinity change within a zincoprotein, and (iii) change in extractability of a zincoprotein. We can say with certainty that the activity of a zincoprotein's function is positively correlated with its abundance (for instance, the speed at which a substrate is converted to the product by its respective enzyme depends on the initial concentration of the said enzyme[52]), while a change in a zincoprotein extractability is most likely not even correlated with its biological activity. The affinity towards $Zn^{2+}$ of a zincoprotein is correlated with its function's activity[53]; however, the direction of this correlation might be not universal. Since the activity of zinc-containing proteins is regulated through zinc homeostasis and zinc-dependent regulatory redox mechanism[54], and the differential extractability issue can be mitigated by the selection of appropriate detergent, we can confidently postulate that the biological activity of a zincoprotein will be a function of its abundance level. Therefore a higher abundance of an identified zincoprotein will result in a higher activity performed by this protein and vice versa. In this vein, we performed PANTHER molecular function and biological process analyses on significantly enriched vs significantly depleted zincoproteins after IVC to identify which cellular functions are being upregulated or downregulated, respectively, during sperm capacitation. To our surprise, we found both upregulated and downregulated functions/activities within the same molecular function categories (binding, catalytic activity, and molecular function regulator). This translates into upregulation and downregulation of biological processes within seven common biological process categories (biological regulation, cellular process, localization, locomotion, metabolic process, response to stimulus, and signaling). Based on our recent understanding, sperm capacitation has been linked with sperm activation or awakening that endows spermatozoa with the potential to fertilize an oocyte. The resultant increase of multiple ion channel permeabilities activates signal transduction pathways that trigger sperm surface protein redistribution, acrosomal remodeling, an increase of the mitochondrial metabolic rate, and the hyperactivation of sperm motility[55], justifying the perception that these functions/processes must be amplified or upregulated during capacitation. Our present data suggest that in capacitating spermatozoa, the regulation of molecular functions performed by zincoproteins and their respective biological processes are even more complex, including both upregulation and downregulation. This is paradigm-shifting and to the best of our knowledge has not been previously reported.

Lastly, we cross-referenced the capacitation-related, significantly different zincoproteins with the literature database in search of the function and localization of the zincoproteins specifically in spermatozoa. We were unable to find any reference to 17 identified zincoproteins. However, we found that the three most represented functions/precesses were sperm motility (21%), energy metabolism (15%), and signaling (13%). The localization of the identified zincoproteins was reported in the sperm head (17%), mitochondria (20%), flagellum (19%), and whole spermatozoon (13%), while no localization was previously reported for 31%. To verify our mass spectrometry results, we phenotyped three significantly different zincoproteins in non-capacitated and capacitated spermatozoa. Two of them, CA2 and CCIN are known to contain/bind $Zn^{2+}$. We confirmed that all three zincoproteins undergo some type of modification during capacitation, which is reflected in one or more observed traits; a change in fluorescence signal intensity, a change in protein abundance level, and/or change of localization. The aim of this study was not to perform an in-depth functional study of these three proteins, but rather to validate our proteomic findings as well as to illustrate capacitation-related changes in these abundant sperm proteins. The implications of the three phenotyped proteins in sperm physiology are discussed in the Supplementary Discussion. Further studies of the identified zincoproteins that change significantly during capacitation and have not been previously reported in spermatozoa are required to understand the nature and implications of their change during this event. Similarly, further attention is required to the mechanisms that orchestrate the changes of zincoproteins during IVC, such as but not limited to cholesterol efflux, activation of protein kinase A, the 26S proteasome activity, and the membrane potential changes of plasma and mitochondrial membranes.

Our study explores the sperm zincoproteome and its implications in porcine sperm capacitation. Our findings further highlight the importance of $Zn^{2+}$ homeostasis in reproduction and offer new avenues in semen handling and processing. Furthermore, our uncovering of zincoproteins that are involved in both sperm physiology and Huntington's and Parkinson's disease pathways predisposes sperm cells to be a potential model for studying these neurodegenerative diseases and exploring the purported link between somatic disease and reproductive system dysfunction.

## Methods

**Antibodies and reagents**. For WB, the following reagents were purchased: Halt^TM Protease and Phosphatase inhibitor cocktail, EDTA free (cat # 78443) was purchased from ThermoFisher Scientific, Rockford, IL, USA. NuPAGE^TM 4–12 % Bis-Tris gel (cat# NP0329BOX) and Novex® Sharp Pre-stained Protein Standard were purchased from Invitrogen, Carlsbad, CA. Bradford protein assay dye (cat # 5000006) was purchased from Bio-Rad, Hercules, CA. PVDF Immobilon Transfer Membrane and Luminata Crescendo Western HRP Substrate were bought from Millipore Sigma, Burlington, MA, USA. CHAPS (cat # 17131401) was purchased from Cytiva Marlborough, MA. Sequencing grade modified trypsin (cat # V5111a) was purchased from Promega Corporation, Madison, WI, USA. All other chemicals used in this study were purchased from Sigma-Aldrich, St. Louis, MO, USA.

For indirect immunofluorescence, image-based flow cytometry, and WB: rabbit polyclonal anti-CA2 (cat # PA5-78897) and rabbit polyclonal anti-CCDC39 (cat # PA5-57243) antibodies were purchased from Invitrogen, Carlsbad, CA. Rabbit polyclonal antibody against CCIN, anti-PT60 was made in house[56,57]. Goat anti-rabbit IgG (cat # 31460), HRP secondary antibody was purchased from ThermoFisher Scientific, Rockford, IL, USA. Goat anti-rabbit IgG, FITC (cat # 62-6111) was purchased from Zymed, San Francisco, CA. Lectin PNA, AF647 (cat # L32460), and DAPI (cat # D1306) were purchased from Invitrogen, Carlsbad, CA.

**Compliance with ethical standards**. Ethical approval: all applicable international, national, and/or institutional guidelines for the care and use of animals were followed. All studies involving vertebrate animals were completed under the strict guidance of an Animal Care and Use protocol, approved by the Animal Care and Use Committee (ACUC) of the University of Missouri. This article does not contain any studies with human participants performed by any of the authors.

**Semen processing and in vitro sperm capacitation**. All studies involving vertebrate animals were completed under the strict guidance of an Animal Care and

Use protocol approved by the Animal Care and Use Committee (ACUC) of the University of Missouri. Fresh boar spermatozoa were collected weekly from a healthy, non-transgenic fertile boar ($n = 1$) used for routine in vitro fertilization trials with high blastocyst yield, and the pre-sperm rich fraction ($n = 10$) was used for the study purposes. The concentration, morphology, and motility of ejaculates were evaluated by conventional semen analysis methods under a light microscope. Sperm concentration was measured by hemocytometer (ThermoFisher Scientific) and ranged from 300 to 500 million/mL; only ejaculates with >80% motile spermatozoa and <20% morphological abnormalities were used for the study. Collections contaminated with urine were discarded. The pre-sperm rich fraction was identified as the first clear fluids, preceding the milkier sperm-rich fraction, and was 5–12 mL. The fractions were free of contaminants other than the expected minimal content of cytoplasmic droplets, thus not necessitating gradient purification. Spermatozoa were separated from seminal plasma by centrifugation (2000 RPM ~ $400 \times g$, 10 min; IEC Centra CL2, ThermoFisher Scientific). Spermatozoa were washed with pre-warmed HEPES buffered Tyrode lactate medium supplemented with polyvinyl alcohol (TL-HEPES-PVA), containing 10 mM sodium lactate, 0.2 mM sodium pyruvate, 2 mM $NaHCO_3$, 2 mM $CaCl_2$, 0.5 mM $MgCl_2$ and 0.01% (w/v) polyvinyl alcohol (PVA); pH = 7.4, $t = 37\,°C$. After the final wash, the sperm pellet was split into halves; the first half was used as a non-capacitated sperm sample and directly processed as described below, while the second half was in vitro capacitated (IVC, 4 h, 37 °C, 5% $CO_2$) as described previously[58]. Spermatozoa after IVC were washed from BSA and used as the capacitated sperm sample. Both non-capacitated and IVC sperm samples were used for protein extraction and flow cytometric studies.

**Immobilized Metal Ion Affinity Chromatography**. The IMAC protocol was adapted from[59,60]. Approximately 500 million non-capacitated and IVC spermatozoa were washed, and resuspended in binding buffer (50 mM Sodium Phosphate, pH = 7.8, 300 mM NaCl) supplemented with 1 mM DTT, 1% (w/v) CHAPS, and protease and phosphatase inhibitors without EDTA, and sonicated on ice with three 20 s bursts, 17% amplitude, and 20 s between each burst. Spermatozoa were then incubated on ice for 30 min with periodical vortexing every 5 min. Following centrifugation at $13,000 \times g$, 15 min, and 4 °C, supernatants and sperm pellets were stored at -25 °C. Zinc chelating resin (G-Biosciences, cat. #786-287) was washed and equilibrated with the binding buffer containing 1 mM DTT, and 1% CHAPS and protein extracts were loaded onto the column and mixed by end-over-end rotation for 1 h, at 4 °C. The resin was pelleted, and the non-binding fraction was saved for further experiments. Resin with bound zincoproteins was washed three times with the binding buffer supplemented with 1 mM DTT and 0.5% (v/v) Triton X-100 (TrX-100) to remove the non-specific binding. Proteins were eluted by a drop of pH with 100 mM sodium acetate buffer, pH = 4.4 (10 minutes, end-over-end rotation at the ambient temperature). Eluates were collected and 4 volumes of acetone were added to precipitate the zincoproteins for proteomic and WB analyses. IMAC protocol was replicated six times in total to ensure robustness and repeatability.

**Proteomics**. IMAC isolated, acetone precipitated zincoprotein pellets were washed with 80% cold (4 °C) acetone twice. The pellet was dissolved in 10 μl of urea buffer (6 M urea, 2 M thiourea, and 100 mM ammonium bicarbonate). The solubilized protein was reduced by DTT and alkylated by iodoacetamide. Next, trypsin (cat # V5111a, Promega) was added for overnight digestion at 4 °C. The digested peptides were desalted with ZipTip with $C_{18}$ resin (cat # ZTC18M096, Millipore Sigma), lyophilized, and resuspended in 10 μL of 5/0.1 % acetonitrile/formic acid. 1 μL of the peptide solution was loaded on a $C_{18}$ column (20 cm×75 μm 1.7 μm) of nanoElute LC-MS system in tandem connection with timsTOF Pro MS/MS system (LC-MS + MS/MS, Bruker Co.) with a step gradient of acetonitrile at 300nL/min. LC gradient conditions: Initial conditions were 2% B (A: 0.1% formic acid in the water, B: 99.9% acetonitrile, 0.1% formic acid), followed by 20 min ramp to 17% B. 17–25% B over 27 min, 25–37% B over 11 min, a gradient of 37% B to 80% B over 6 min, hold at 80% B for 6 min. The total LC run time was 70 min. MS data were collected over an m/z range of 100 to 1700. During MS/MS data collection, each TIMS cycle included 1 MS + an average of 10 PASEF MS/MS scans. Raw data were searched using PEAKS (version X) with UniprotKB *Sus scrofa* protein database downloaded on Mar 01, 2019, with 88,374 entries. Data search was adjusted for trypsin digestion with initial two missed cleavage sites, fixed modification by carbamidomethylation, and variable modification by methionine oxidation and pyroglutamate formation at peptide N-terminal glutamic acid. The precursor ion mass error tolerance was ±20 ppm and the MS/MS fragment ion mass error tolerance was ±0.1 Da. For the identification of proteins, the following criteria were used: ≥1 unique peptide and ≥2 peptides per protein; the false discovery rate was set to 1%. The proteomic study was done in one pilot run and three additional full replicates. Each run used a new sperm collection. Identified proteins with measured spectral counts were combined, and the protein spectrum number was normalized based on the total spectrum number and internal control protein (sperm head acrosin) in each sample. The pilot replicate was excluded from the data analysis due to the unsatisfactory number of total proteins identified, which was the result of a relatively short LC run (30 min) as opposed to the 70 minutes used in the working replicates. The data from three working replicates were used for data analysis. Results are presented in the Excel spreadsheet as Supplementary Data 1.

**Immunofluorescence**. Non-capacitated and IVC capacitated spermatozoa were either fixed in 2.0% (v/v) formaldehyde in phosphate-buffered saline (PBS; 137 mM NaCl, 2.7 mM KCl, 10 mM $Na_2HPO_4$, 1.8 mM $KH_2PO_4$, pH = 7.4), and permeabilized in 0.1% (v/v) TrX-100 in PBS (PBST) or fixed and permeabilized in 90% ice-cold (-25 °C) methanol. Approximately 50 million spermatozoa were blocked with PBST supplemented with 5% (v/v) normal goat serum (NGS). All the antibodies used for flow cytometric studies were previously characterized. Primary antibodies were used as follows: anti-carbonic anhydrase 2 (CA2, 1:80 dilution; PA5-78897, Invitrogen), anti-coiled coil domain 39 (CCDC39, 1:100 dilution, PA5-57243, Invitrogen), and anti-calicin [CCIN/PT60; 1:50 dilution;[56]] all diluted in PBST supplemented with 5% (v/v) NGS. Primary antibodies were added to sperm sample tubes and incubated overnight at 4 °C. For the primary control, non-immune rabbit sera of comparable globulin concentrations were used instead of primary antibodies and processed in the same fashion. The following morning, spermatozoa were washed twice with PBST with 1% (v/v) NGS, and appropriate species-specific secondary antibody, goat anti-rabbit conjugated to fluorescein isothiocyanate (GAR-FITC, 1:200 dilution, Zymed) in PBST with 1% (v/v) NGS was added and allowed to incubate for 40 min at room temperature. For acrosome integrity assessment, peanut agglutinin conjugated to Alexa Fluor 647 (PNA-AF647, 1:2000 dilution; Molecular Probes) was used, and 4′,6-Diamidino-2-Phenylindole Dilactate (DAPI), a DNA stain (1:1500 dilution; Molecular Probes) was used as a reference and nuclear contrast stain. Both PNA-AF647 and DAPI were mixed and coincubated with secondary antibodies. After incubation with secondary antibodies, spermatozoa were washed twice with PBST with 1% (v/v) NGS. For the secondary control; secondary antibodies were omitted as well as DAPI and PNA-AF647 and processed in the same fashion. All steps were accomplished in suspensions.

**Epifluorescence microscopy**. The fluorescently labeled samples were mounted on microscope slides with VectaShield (Vector Laboratories, Inc., Burlingame, CA, #H-1000) and imaged using a Nikon Eclipse 800 microscope (Nikon Instruments Inc.) with Retiga QI-R6 camera (Teledyne QImaging, Surrey, BC, Canada) operated by MetaMorph 7.10.2.240. software (Molecular Devices, San Jose, CA). Images were adjusted for contrast and brightness in Adobe Photoshop 2020 (Adobe Systems, Mountain View, CA, USA), to match the fluorescence intensities viewed through the microscope eyepieces.

**Image-based flow cytometry**. The fluorescently labeled samples were measured with an Amnis FlowSight Imaging Flow Cytometer (AMNIS Luminex Corporation, Austin, TX) as described previously[61]. The instrument was fitted with a ×20 microscope objective (numerical aperture of 0.9) with an imaging rate of up to 2000 events per sec. The sheath fluid was PBS (without $Ca^{2+}$ or $Mg^{2+}$). The flow-core diameter and speed were 10 μm and 66 mm per second, respectively. The raw image data were acquired using INSPIRE® software (AMNIS Luminex Corporation, Austin, TX). To produce the highest resolution, the camera setting was at 1.0 μm per pixel of the charge-coupled device. Samples were analyzed using four lasers concomitantly: a 405-nm line (10 mW), 488-nm line (10 mW), 642-nm line (20 mW) a 785-nm line (50 mW, side scatter), and two LEDs (32.57 mW and 19.30 mW respectively). Signals were observed in the following channels: channels 1 and 9 – brightfield, channel 2 – green fluorescence (AF488, 505–560 nm), channel 6 (SSC), channel 7 – blue fluorescence (DAPI, 435–505 nm), and channel 11 – infrared fluorescence (AF647, 642–745 nm). A total of 10,000 events were collected per sample, and data were analyzed using IDEAS® software (Version 6.2.189.0; AMNIS Luminex Corporation, Austin, TX). A focused, single-cell population gate (Fig. S15) was used for histogram display of mean pixel intensities by frequency for collected channels. Intensity histograms of individual channels were then used for drawing regions of subpopulations with varying intensity levels and visual confirmation. The intensity of DAPI was used for histogram normalization among non-capacitated and IVC spermatozoa. Fluorescently positive debris was excluded by applying masks, as well as calculating the fluorescence intensity of the post-acrosomal segment. The primary control of normal rabbit serum is shown in[31], and the secondary control is presented in Fig. S16.

**Protein isolations from spermatozoa and tissues**. The processing of an ejaculate to obtain spermatozoa and seminal plasma as well as sperm IVC was described earlier. Boar kidneys and testes were obtained from a humanely slaughtered, fertile adult animal at the National Swine Resource and Research Center, University of Missouri. Approximately 100 mg of the tissue was collected from the organs, homogenized on ice, mixed with 2X LDS loading buffer supplemented with proteasomal inhibitors, and incubated for two hours at 4 °C with end-over-end rotation. After the incubation, the suspensions were spun at $13,000 \times g$, 4 °C, for 15 minutes. Supernatants were transferred to new Eppendorf tubes and before the PAGE loading, the solution was adjusted to 1X LDS buffer with 2.5% (v/v) 2-mercaptoethanol.

The isolation protocol for the perinuclear theca proteins was adapted from[37]. Briefly, 100 million PBS washed, non-capacitated or IVC spermatozoa were resuspended in 1 mL of cold (4 °C) PBS supplemented with proteasomal inhibitors and were sonicated on ice by six 20 s bursts to remove plasma, and intraacrosomal membranes as well as the acrosomal matrix (40% amplitude, 30 s pauses between

bursts; Branson Ultrasonics). Spermatozoa were spun after sonication ($1000 \times g$, 5 mins, 4 °C). For the parallel extractions, the pellet was resuspended in 2X LDS (described below), or 100 mM NaOH aqueous solution, and incubated at 4 °C, overnight with end-over-end rocking. For the sequential extractions, the pellet was resuspended in 0.2% (v/v) TrX-100 in PBS with protease inhibitors, and incubated for an hour at RT with end-over-end rotation. After the incubation, the sperm suspension was spun at $2500 \times g$, 10 min, 4 °C. The supernatant was saved for later analysis. In the second step, the sperm pellet was resuspended in 1 M KCl in PBS and incubated for an hour at RT with end-over-end rotation. After the incubation, the suspension was spun at $2500 \times g$, 10 min, 4 °C. The supernatant was desalted by the addition of 4 volumes of acetone resulting in salt precipitation. The salt was pelleted, the supernatant was transferred to a new Eppendorf tube, and was left to incubate at -25 °C overnight. The sperm pellet after the second incubation was resuspended in an aqueous, 100 mM NaOH solution, and incubated at 4 °C, overnight with end-over-end rocking. After the overnight incubation for both sequential and parallel isolations, suspensions were spun at $2500 \times g$ for 10 min at 4 °C. Extracts were transferred to new Eppendorf tubes, and 100 mM NaOH extracts were neutralized with 100 mM Sodium Phosphate buffer (pH = 7). All the other extracts/pellets were adjusted so that the final solution contained 1X LDS buffer with 2.5% (v/v) 2-mercaptoethanol. A 20 million sperm equivalent and all precipitated proteins from the second step of sequential isolation were loaded per lane. Proteins were probed with the anti-CCIN antibody as described below.

**Western blotting**. Acetone precipitated zincoproteins from non-capacitated and IVC spermatozoa were resuspended in lithium dodecyl sulfate (LDS) loading buffer (106 mM Tris HCl, 141 mM Tris base, 2% (w/v) LDS, 10% (w/v) glycerol, 0.51 mM (0.75% w/v) EDTA, 0.22 mM (0.075% w/v) Coomassie Brilliant Blue G250, 0.175 mM (0.025% w/v) Phenol Red, pH = 8.5) supplemented with 2.5% (v/v) 2-mercaptoethanol, and protease and phosphatase inhibitor cocktail. Protein extracts of non-capacitated and IVC spermatozoa before and after IMAC were mixed in the appropriate ratio with 4X reducing LDS buffer, and the non-capacitated and IVC sperm pellets were resuspended in 1X reducing LDS buffer. Samples were incubated for 10 minutes at 70 °C, before loading on a gel.

For polyacrylamide gel electrophoresis (PAGE), a NuPAGE™ electrophoresis system was used (Invitrogen, Carlsbad, CA). A total of 30 µg protein (concentration estimated by BCA assay, cat #23227, ThermoFisher Scientific) was loaded per single lane to probe the input and the non-binding fractions; an equivalent of 20 million spermatozoa was loaded for sperm pellet extracts, and 1/3 of precipitated zincoproteins from single IMAC was loaded per well. PAGE was carried out on NuPAGE™ 4–12% Bis-Tris gel (cat# NP0329BOX, Invitrogen, Carlsbad, CA) using TRIS-MOPS SDS Running Buffer [50 mM Tris Base, 50 mM 3-(N-morpholino)propanesulfonic acid (MOPS), 0.1% (w/v) SDS, 1 mM EDTA, pH = 7.7]. Anode buffer (the top one) was supplemented with 5 mM NaHSO₃ to prevent reoxidizing of disulfide bonds. The molecular masses of separated proteins were estimated using Novex® Sharp Pre-stained Protein Standard (cat # LC5800, Invitrogen, Carlsbad, CA) run in parallel. PAGE was carried out for 5 minutes at 80 V to let the samples delve into the gel and then for another 60–70 mins at 160 V. The power was limited to 20 W. After PAGE, proteins were electrotransferred onto a PVDF Immobilon Transfer Membrane (Millipore Sigma) using an Owl wet transfer system (Thermo Fisher Scientific) at 300 mA for 90 mins for immunodetection, using Bis-Tris-Bicine transfer buffer (25 mM Bis-Tris base, 25 mM Bicine, 1 mM EDTA, pH = 7.2) supplemented with 10% (v/v) methanol per membrane, and 2.5 mM NaHSO₃.

The PVDF membrane with the transferred proteins was blocked with 10% (w/v) non-fat milk (NFM) in Tris-buffered saline (TBS; 50 mM TRIS-HCl, pH = 7.4, 137 mM NaCl) with 0.05% (v/v) Tween 20 (TBST; Sigma-Aldrich) and incubated with the primary antibody overnight. Primary antibodies used were as follows: anti-carbonic anhydrase 2 (CA2, 1:4000 dilution, PA5–78897, Invitrogen), anti-coiled coil domain 39 (CCDC39, 1:2000 dilution, PA5-57243, Invitrogen), and anti-calicin (CCIN/PT60; 1:4000 dilution[56]) all diluted in TBST supplemented with 5% (w/v) NFM. The next day, the membrane was incubated for 40 min with an appropriate species-specific secondary antibody, such as the HRP-conjugated goat anti-rabbit antibody (GAR-IgG-HRP, 1:10,000 dilution; cat # 31460 Invitrogen). The membrane was reacted with chemiluminescent substrate (Luminata Crescendo Western HRP Substrate; Millipore Sigma) and the blot was screened with ChemiDoc Touch Imaging System (Bio-Rad, Hercules, CA, USA), to record the protein bands. The image was analyzed by Image Lab Touch Software (Bio-Rad, Hercules, CA, USA). Where not specified, procedures were carried out at room temperature. Membranes were stained with CBB R-250 after chemiluminescence detection for protein load control.

**Zinc binding site prediction**. To run prediction for zinc-binding sites on the candidate zincoproteins, we obtained the raw protein sequences by passing each accession number as a parameter in each Application Programming Interface (API) call made using the EMBL-EBI dbfetch tool[62]. We then ran a prediction for zinc-binding sites on these sequences using our in-house predictor ZinCaps[63] based on the Capsule Network architecture. For each candidate binding residue, i.e., Cys, His, Glu, and Asp (CHED), we report the prediction probability. Binding residues with a probability of 0.5 and higher were taken into account for the

purpose of this study. All prediction sites including their respective probabilities are included in the Supplementary Data 2.

**Statistics and reproducibility**. For the proteomics, one trial replicate (for optimization of pull-down conditions) and three working replicates were performed. Protein spectral counts from the three working replicates were combined, the protein spectrum number was normalized based on the total spectrum number and internal control protein (acrosin) in each sample, and a two-tailed $t$-test with $\alpha = 0.05$ for each identified protein was performed. For IBFC and WB, five and eight replicates were conducted, respectively. Paired, two-tailed t-test was performed on IBFC, and a two-sample, two-tailed $t$-test was performed on WB data with $\alpha = 0.05$ for both. A $P$-value <0.05 was considered statistically significant.

All experiments in this article have been replicated with authenticated reagents, appropriate controls, and consistent results between replicates. The study was conducted on a wild-type, non-transgenic boar ($n = 1$), and the proteomic study was conducted in 1 pilot and 3 working replicates. The results from 3 technical replicates were used for bioinformatic analyses. The IBFC and WB phenotyping experiments were replicated 5 times and 8 times, respectively. Readers are strongly encouraged to peruse Methods section and contact the corresponding author should concerns arise about the reproducibility.

**Reporting summary**. Further information on research design is available in the Nature Research Reporting Summary linked to this article.

## Data availability

The MS proteomics data in this paper have been deposited in the ProteomeXchange Consortium (http://proteomecentral.proteomexchange.org) via the MassIVE partner repository with the data set identifier MSV000089238. All source data are included as supplementary data. All other data are available upon reasonable request. Correspondence and requests for materials should be addressed to M.Z. or P.S.

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

## Acknowledgements

This project was supported by USDA National Institute of Food and Agriculture, Agriculture and Food Research Initiative Competitive Grants no. grants 2020-67015-31017 (P.S.), 2021-67015-33404 (P.S.), and 2019-67012-29714 (K.K.), University of Missouri CAFNR Joy of Discovery Seed Grant award (P.S and M.Z.), and NIH grant R35-GM126985 (C.E. and D.X.). We thank the Gehrke Proteomics Center at the University of Missouri, Columbia for mass spectrometry services; National Swine Resource and Research Center (NSRRC) staff for boar semen collection; Ms. Kathy Craighead for editorial and administrative assistance, and Ms. Miriam Sutovsky for technical and managerial assistance.

## Author contributions

P.S., D.X., K.K. and M.Z. study conceptualization; M.Z. wet experiments; K.K., S.S., and C.E. bioinformatics; K.K., M.Z. formal analysis; P.S., R.O., D.X., K.K., and M.Z. manuscript drafting; P.S., K.K., and D.X. funding acquisition. All authors have read and agreed to the published version of the manuscript.

## Competing interests

There is no conflict of interest that could be perceived as prejudicing the impartiality of the research reported. It is disclosed that P.S. and K.K. are the founders and owners of AndroLabb LLC, Columbia, MO, a biotech startup involved with sperm processing and sperm quality diagnostics, which did not play any role in the trials described herein.
