## [Peer Review File · Communications Biology]

Reviewers' comments:

Reviewer #1 (Remarks to the Author):

Comments to the Author

Summary

The submitted material is a novel study to shed light on potential roles of zinc in porcine sperm function. Special emphasis is placed on the shifting role of zincoproteins after capacitation complemented with sequencing and bioinformatic approaches to determine plausible roles in sperm function. The methodology employed is highly appropriate for the studies conducted with concise and rigorous efforts made towards experimental design and data transparency.

As stated in Line 250, the aim of the study was to validate proteomic findings and capacitation-related changes of proteins without performing in-depth functional analyses. The reviewer finds that this statement is accurate but needs to be reflected in the title and introduction. At present, the data presented from bioinformatics do not exclusively test functions of "sperm binding, motility and metabolic modulation" but instead are inferred from various biomedical journals as known functions in various cell types. In addition, without comparative cross-species analyses of these experiments and n=1 boar, it is difficult to conclude if the outputs are indeed representative of "mammalian sperm capacitation" rather than unique to porcine.

Specific Comments

- It appears that one boar was used throughout the entire experiment to characterize Zincoprotein abundance in capacitated and uncapacitated sperm. Why was semen from multiple boars not pooled to increase biological specie representation when performing seminal experiments to describe novel data? Would not the results grossly characterizing zinc function in mammalian sperm be potentially under or over-represented by one individual animal?
- Please consider changing title to "porcine" instead of "mammalian" as species differences likely exist but have not been explored within this work. The inherent species differences are well emphasized in supplemental material (p 2-4) with specific attention regarding localization with implications to sperm function.
- The reviewer appreciates the complexity, thoroughness and rigor achieved of data submission throughout the manuscript and can imagine the difficult task of appropriately selecting, displaying and conveying pertinent information. Despite this great effort, some re-organizing needs to be accomplished to appropriately display and convey data to readers as indicated below:
- The manuscript includes four figures, three of which are doughnut representation of PANTHER functions. Some of these data would be better represented by those that were submitted as supplementary material.
- Figure 1: Despite zooming to 300%, some of these data are still not readable. The reviewer finds that Graphic Panther Analysis located in Table S2 much more useful in terms of clear and concise representation of data. There are simply too many terms to display clearly in doughnut charts. If doughnut charts are desired, the reviewer recommends limiting the amount of information displayed to make them legible and then adding figures from the Graphic Panther Analyses Views to the main manuscript rather than as supplementary data. In fact, the reviewer greatly prefers these well-displayed analyses with clickable links.
- Figure 2: Please define NS. Please explain why P-value was unadjusted. Consider enumerating each category (Red, Blue, Green, Gray) as a representation of total number identified and/or percentage.
- Figure 3: Many of the same comments as Figure 1, although easier to read the font in A,C but not legible in B,D. Again, these images would be better displayed as graphs by moving some of the supplementary files into the main text. If the doughnut representations are desired, I suggest

labelling A,C as "Molecular Functions" and B,D as "Biological Process". An explanation of "high" and "low level" needs to be made for those not familiar. These figures cannot stand alone as is due to legibility.

- Figure 4: Perhaps consider including text "Protein Function" and "Localization" above A & B, respectively for ease of interpretation.

- The Extended Data: These data appear to be among the most novel. The manuscript would benefit from a more fluid approach to incorporating these images and descriptions without having to float back and forth between supplementary information. The imbedded images at the top left corners of Extended Data 2 are almost illegible in A,D except at 250%. Without flipping back to supplementary information, it is difficult to know if represented images are for capacitated and uncapacitated, especially since PNA staining is apparent and methods to describe permeabilization (or lack thereof) need to be hunted down in the methodology section. The reviewer certainly appreciates the complexity of trying to display this data but some effort needs to be made for more fluid reading or perhaps incorporation in a separate manuscript. It also appears that the WB is difficult to interpret, presumably for technical reasons (figure E). If significance in abundance occurs from WB, they should be made clear in legend and data ($P < 0.05$).

Specific comments to the Manuscript

- Title: the word "mammalian" should be replaced with "porcine" due to many inherent inter-species differences in sperm proteins described but specifically unexplored in relation to zincoproteins herein.
- Prior to the Methods section, there is NO mention (unless the reviewer missed it) of the species used to perform these analyses. The first mention is a sus scrofa reference in methods section. Porcine needs to be included within main text.
- Abstract: what new avenues are offered for semen handling, assisted reproductive therapy and livestock breeding?
- Line 131: indicate that "changing" refers to abundance etc.
- Line 43: Is the word "Introduction" missing?
- The words "reviewed in" are used throughout the manuscript and are not necessary
- Line 51: Please describe what "free Zn" refers to. Intracytosolic etc?
- Line 53: Please describe if "chelated" refers to the plasma membrane or extracellular environment although mentioned at beginning of Suppl data.
- Line 41-53: would benefit from a brief description of where Zn is abundant throughout reproductive processes that involve sperm for broader relevance and journal scope
- Line 57: does "permissiveness" describe activation? Please clarify
- Line 70: does "high concentration of Zn" refer to extracellular?
- Line 72: what species?
- Line 87: Please indicate that the objective was to isolate these proteins from sperm
- The word ejaculated is confusing throughout because both capacitated and un(non)capacitated sperm are derived from ejaculated sperm. The reviewer suggests removing this terminology and simplifying with capacitated and non(un)capacitated
- Line 94-98: Very brief introduction is necessary for specie, number of males used, number of ejaculates and whether fresh, chilled or frozen-thawed and if ejaculates were always split
- Line 100-101: please describe why some replicates were removed from analyses objectively if not included in Methods
- Line 152: remove semicolon
- Line 156-160: please restructure for flow
- Line 160: These 3 proteins seem to be important but results are not mentioned, likely due to restriction of journal scope etc. The reviewer suggests either elaborating on these novel findings or eliminating to submit in a separate manuscript.
- Line 173 (cite suppl. Data)
- Line 175 (first mention of species in manuscript unless reviewer missed it) needs to be made clear in intro
- Line 186-190: restructure run-on sentence for clarity and flow
- Line 191: How would a block to polyspermy and sperm binding to oviductal epithelium lead the authors to conclude that Zn can reversibly alter sperm motility when both interactions primarily

depend on plasma membrane function?

- How do the authors suggest that a transcriptionally and translationally quiescent protamine compact cell can model neurodegenerative diseases? Testicular function and sperm quality may indirectly be related to transgenic models. Without further explanation, the implications may be beyond the scope of results.
- Line 216: please provide citation to support
- Line 221: Do low abundant zincoproteins have less biological function? Not clear
- Line 256: How do we know these proteins are conserved across mammals? Study was not based on Zn homeostasis, semen handling or processing. Please help to complete the link between a germ cell and neurodegenerative disease model.
- Line 414: Please indicate which methods "Antibodies and Reagents" were purchased for. (reader does not know that WB) was being performed so difficult to read and follow
- Line 451: Please briefly list capacitating conditions
- Line 453: What are incubation conditions for NC sperm?
- Line 467: Secondary binding of what?
- Line 511: primary and secondary controls?
- Line 520: was staining accomplished in suspension or mounted (methods)
- Line 521: Please include EX/EC parameters for each secondary, intensity and imaging instrumentation (LED epifluorescence etc), how many sperm analyzed and number of reps
- Justification for choosing probability 0.5?
- Line 631: Move to methods
- Line 702: Please make x-axis more clear: e.g. make clear what 1-5, 6-10 etc means (# amino acids?)

Supplemental Data

- FigS3: Suggest to label columns as non-capacitated and capacitated for ease
 - o Are these non-permeabilized sperm? (include in figure legend)
 - o Were appropriate primary and secondary controls used to identify specific vs. non-specific binding
 - o A: difficulty seeing red in the AR of Capacitated sperm (is it there)?
 - o EF- don't see staining in PP of F
- FigS4: Why is there discrepancy in y-axis relative intensity among all 4 replicates? Can you please explain the bimodal distribution of reps 3 and 5
- Fig S5 lacks sufficient description in legend to tie experimental objective to results depicted. Text is presented on P2 of supplemental material but visualization of band(s) of interest including loading controls is difficult to interpret. Perhaps a square around area of interest to guide reader along with simple mean relative intensity graphic would be helpful.
- Fig S6: Please clarify that the 5 different CCIN populations are not discrete proteins (isoforms) but instead refer to localization patterns
- Fig S9: why the discrepancy in detection of clean bands between reps 1-4 and 5-8? How are the authors able to determine relative intensity using reps 5-8 when bands run together around 50-60kD, yet single bands appear in 1-4 around 60kD? Perhaps difficulty in solubility?
- Fig S10: A and B need to move above respected isolation techniques rather than on the sides of the figure. KCL seems to pull down much more protein in Capacitated but total protein is less, which indicates that either CCIN abundance is higher or more has gained additional access to proteins with non-specific binding. How do the authors image these bands for quantification given the likelihood of non-specificity around 60KD?
- Fig S11: Reps 2 and 4 seems somewhat consistent but 5 appears to be an outlier. Was rep 5 included in analyses?

Reviewer #2 (Remarks to the Author):

Summary of manuscript:

The manuscript contains reliable data on analysis of Zinc interacting proteins. Proteins were isolated using IMAC with immobilized Zn²⁺. While Zn²⁺ chelation before IMAC resulted in no pull

down of these proteins (a good way for showing that these proteins had Zn²⁺ dependent interaction).

One part of the manuscript deals with proteomic analysis of the Zinc protein interactome (1740 HUGO annotated proteins). These proteins were classified into molecular functionality or in which biological processes and pathways they are involved.

The second part deals with capacitation induced changes of the zinc protein interactome. Indeed 102 proteins showed significant changes. Three most observed functions or processes that showed changes were related to sperm motility, energy metabolism and signaling (all three known to change indeed during sperm capacitation).

Three significant changed zinc interacting proteins were phenotyped (an showed capacitation dependent changes in immune fluorescent signal intensity, or in immune fluorescent localization, or a change in protein abundance.

Of interest is also that Huntington's and Parkinson's disease pathways are involved in sperm processes studied. The authors hypothesize that sperm could be a relevant model for studying these diseases.

General points:

It is unusual to me to see a reference to previous work in the abstract. Is this allowed in the format of Comm. Biol.?

Sections: I am used to manuscripts with sections heading Introduction (for line 43-91) which is missing, Results line 93-162) which is also lacking. Is the format for Comm. Biol. different from this?

The authors mentioned in line 250-252 that this study lacks to link the observation to functionality, but that their aim was to present a list of zinc interacting proteins and capacitation dependent changes thereof. The lack of functional data is indeed a shortcoming. I think that the lack of functional data is acceptable for this manuscript but in this section at least some ideas about how to study this shortcoming. For instance are the Zinc interacting protein changes dependent on protein kinase A activation or cholesterol efflux or on membrane potential (the latter can be tested on plasma membrane level or in the mitochondria). So I would request some lines in which the authors give the reader a hint in how to experimentally study the functionality of changes in zinc interacting proteins in sperm.

Specific points:

page 3 ending in line 91 line 56-64 cholesterol efflux is a process that is slower (hour time scale) than the activation of SACY (10 minutes time scale). This part must be rephrased accordingly.

References in general: is not in consistent format regarding Capital letters or not sometimes all nouns are capitalized (for instance ref 1,2,3) and many times they are not (for instance 4, 5,6). Idem sometimes the nouns are spelled in full (for instance ref 3, 4) while sometimes they are abbreviated (for instance ref 1,2).

Ref 17, 27, 28: these citations contain the city and country of the journal *Reproduction*. I think that part should be skipped.

Response to review COMMSBIO-21-2018-T

We thank the editorial board and both reviewers for thoughtful comments, which we followed closely as we revised our manuscript.

Reviewer #1 (Remarks to the Author):

Summary

The submitted material is a novel study to shed light on potential roles of zinc in porcine sperm function. Special emphasis is placed on the shifting role of zincoproteins after capacitation complemented with sequencing and bioinformatic approaches to determine plausible roles in sperm function. The methodology employed is highly appropriate for the studies conducted with concise and rigorous efforts made towards experimental design and data transparency.

Response: We thank reviewer 1 for recognizing our efforts and their positive feedback.

As stated in Line 250, the aim of the study was to validate proteomic findings and capacitation-related changes of proteins without performing in-depth functional analyses. The reviewer finds that this statement is accurate but needs to be reflected in the title and introduction. At present, the data presented from bioinformatics do not exclusively test functions of “sperm binding, motility and metabolic modulation” but instead are inferred from various biomedical journals as known functions in various cell types. In addition, without comparative cross-species analyses of these experiments and n=1 boar, it is difficult to conclude if the outputs are indeed representative of “mammalian sperm capacitation” rather than unique to porcine.

Response: We apologize for the confusion; L 250 was in regard to the 3 phenotyped proteins. The aim of the study was to identify all zinc interacting/binding proteins from non-capacitated and capacitated spermatozoa and to find zincoproteins that change during capacitation. For the significantly different zincoproteins after capacitation, we performed a literature search (tab S3) primarily focused on sperm papers to give us as accurate picture about protein function and localization in spermatozoa as possible. There were indeed some proteins that have not been reported in spermatozoa, in which case we assumed their function either on the protein family relatedness to proteins reported in sperm; or known function in other cell types. The same goes for the localization of said proteins. With no dedicated sperm proteome database, we believe that this is the closest approximation of the function and localization in the spermatozoa of identified, significantly different proteins after capacitation. We phenotyped 3 significantly different proteins to validate our mass spectrometry results. In order to make this clear, we reformulated the objectives of the study (L88-91 as well as L250 of the initially submitted MS). We agree with the reviewer that a study done in boar does not necessarily reflect other mammalian species closely; therefore, we changed “mammalian” to “porcine” in the title.

Specific Comments

- It appears that one boar was used throughout the entire experiment to characterize Zincoprotein

abundance in capacitated and uncapacitated sperm. Why was semen from multiple boars not pooled to increase biological specie representation when performing seminal experiments to describe novel data? Would not the results grossly characterizing zinc function in mammalian sperm be potentially under or over-represented by one individual animal?

Response: This is correct, we only used one boar from which we collected four ejaculates at different times and thus performed four replicates. The reason being is simply because, at the time of this experiment, we had only had one WT boar at our disposal. This was an adult, healthy boar of proven fertility, used for IVF at the National Swine Research Resource facility supplying semen for our studies; it is a routine practice at this NIH-supported facility to train and collect one boar only at a time for artificial insemination and in vitro embryo production. We could obtain semen from other boars but their in vitro and in vivo fertility would not be proven. We do recognize and acknowledge that the zinc function in porcine spermatozoa could be somewhat under or over-represented by one individual; on the other hand, due to a long time since domestication, the pig possesses lower heterozygosity, higher inbreeding coefficient, and larger linkage disequilibrium when compared with human and cattle (doi: [10.2174/138920211795564386](https://doi.org/10.2174/138920211795564386)).

- Please consider changing title to “porcine” instead of “mammalian” as species differences likely exist but have not been explored within this work. The inherent species differences are well emphasized in supplemental material (p 2-4) with specific attention regarding localization with implications to sperm function.

Response: We have changed the title according to the reviewer's suggestion.

- The reviewer appreciates the complexity, thoroughness, and rigor achieved of data submission throughout the manuscript and can imagine the difficult task of appropriately selecting, displaying, and conveying pertinent information. Despite this great effort, some re-organizing needs to be accomplished to appropriately display and convey data to readers as indicated below:

Response: We thank the reviewer for their positive, and constructive critique; which we followed to the letter.

- The manuscript includes four figures, three of which are doughnut representation of PANTHER functions. Some of these data would be better represented by those that were submitted as supplementary material.

Response: The original MS, submitted to Nature before transfer to Commun. Biol. included 6 figures (4 figures + 2 extended), due to Nature restrictions. Fortunately, Commun. Biol. permits up to 10 displayed items; therefore, we rearranged it as it was in the original MS before the submission to Nature.

- Figure 1: Despite zooming to 300%, some of these data are still not readable. The reviewer finds that Graphic Panther Analysis located in Table S2 much more useful in terms of clear and concise representation of data. There are simply too many terms to display clearly in doughnut charts. If doughnut charts are desired, the reviewer recommends limiting the amount of information displayed to make them legible and then adding figures from the Graphic Panther Analyses Views to the main manuscript rather than as supplementary data. In fact, the reviewer greatly prefers these well-displayed analyses with clickable links.

Response: The relevant figure (now figure 2 after rearrangement) now consists of graphic PANTHER analysis from table S2 with clickable links. Doughnut charts were simplified for clarity and moved to the Supplementary material.

- Figure 2: Please define NS. Please explain why P-value was unadjusted. Consider enumerating each category (Red, Blue, Green, Gray) as a representation of total number identified and/or percentage.

Response: NS is now defined. P-value did not need to be adjusted, since our protein differential abundance measurement is only between before and after 4 hours of capacitation. This means only one comparison is being conducted and there is no multiple hypotheses being tested in this experiment, hence not necessitating the need for adjusted p-value, q-values, or FDR correction.

- Figure 3: Many of the same comments as Figure 1, although easier to read the font in A,C but not legible in B,D. Again, these images would be better displayed as graphs by moving some of the supplementary files into the main text. If the doughnut representations are desired, I suggest labelling A,C as “Molecular Functions” and B,D as “Biological Process”. An explanation of “high” and “low level” needs to be made for those not familiar. These figures cannot stand alone as is due to legibility.

Response: In this case, doughnut charts are desirable to show the up/down-regulation within the categories. The explanations of high and low levels are now included in the respective figure legend.

- Figure 4: Perhaps consider including text “Protein Function” and “Localization” above A & B, respectively for ease of interpretation.

Response: We implemented the changes suggested by the reviewer.

- The Extended Data: These data appear to be among the most novel. The manuscript would benefit from a more fluid approach to incorporating these images and descriptions without having to float back and forth between supplementary information. The embedded images at the top left corners of Extended Data 2 are almost illegible in A,D except at 250%. Without flipping back to supplementary information, it is difficult to know if represented images are for capacitated and uncapacitated, especially since PNA staining is apparent and methods to describe permeabilization (or lack thereof) need to be hunted down in the methodology section. The reviewer certainly appreciates the complexity of trying to display this data but some effort needs to be made for more fluid reading or perhaps incorporation in a separate manuscript. It also appears that the WB is difficult to interpret, presumably for technical reasons (figure E). If significance in abundance occurs from WB, they should be made clear in legend and data ($P < 0.05$).

Response: We agree with the reviewer completely. Extended data were created to comply with the Nature format, where the MS was originally submitted to. Extended data are now part of the manuscript. We also moved the results from the supplemental material to the MS. We, however, left the discussion part relevant to these data in the supplemental material so that the main discussion is not too long. The font size in the embedded images was increased. The images are labeled on top so it is certain whether they are from capacitated or non capacitated spermatozoa (except panel D where they are all from capacitated fraction, this is now labeled as well). Permeabilization, as well as other details, are now included in the figure legend so that the reader does not have to search it in the M&M.

Significance is now made clear in both the figure and the legend.

Specific comments to the Manuscript

- Title: the word “mammalian” should be replaced with “porcine” due to many inherent inter-species differences in sperm proteins described but specifically unexplored in relation to zincoproteins herein.

Response: The title was corrected according to the reviewer's suggestion.

- Prior to the Methods section, there is NO mention (unless the reviewer missed it) of the species used to perform these analyses. The first mention is a sus scrofa reference in methods section. Porcine needs to be included within main text.

Response: Porcine species is now mentioned in the title as well as in the intro. A word “Pig” was added to keywords, too.

- Abstract: what new avenues are offered for semen handling, assisted reproductive therapy and livestock breeding?

Response: Zincoprotein phenotyping and management offers opportunities to improve boar fertility evaluation, increase the viability and fertility of liquid boar semen during distribution for artificial insemination, and reduce sperm number per dose as well as the number of boars trained and kept for semen collection, thus reducing the expense and environmental impact of commercial boar studs.

- Line 31: indicate that “changing” refers to abundance etc.

Response: This is now indicated.

- Line 43: Is the word “Introduction” missing?

Response: This was in accordance with Nature format where the manuscript was originally submitted. It is now included.

- The words “reviewed in” are used throughout the manuscript and are not necessary

Response: The use of this word has been reduced.

- Line 51: Please describe what “free Zn” refers to. Intracytosolic etc?

Response: We apologize; it was supposed to refer to labile zinc (i.e. free/loosely bound/available zinc) as opposed to tightly associated zinc that is unavailable to the cell.

- Line 53: Please describe if “chelated” refers to the plasma membrane or extracellular environment although mentioned at beginning of Suppl data.

Response: Chelated in general, from the whole spermatozoa. In our study, ref #9 where we added TPEN, a specific Zn chelator to live spermatozoa, the zinc signal completely disappeared. After the addition of $ZnCl_2$ the signal was restored, presumably zinc bound to the same compartments as prior to the chelation.

- Line 41-53: would benefit from a brief description of where Zn is abundant throughout reproductive processes that involve sperm for broader relevance and journal scope

Response: This is now included.

- Line 57: does “permissiveness” describe activation? Please clarify

Response: Yes, it does. Permissiveness was changed to activation.

- Line 70: does “high concentration of Zn” refer to extracellular?

Response: Yes indeed; the word “extracellular” has been added.

- Line 72: what species?

Response: In humans and possibly bulls as well. A line has been updated accordingly.

- Line 87: Please indicate that the objective was to isolate these proteins from sperm

Response: This is now indicated. We also indicated that it was porcine spermatozoa as per other suggestions.

- The word ejaculated is confusing throughout because both capacitated and un(non)capacitated sperm are derived from ejaculated sperm. The reviewer suggests removing this terminology and simplifying it with capacitated and non(un)capacitated

Response: The word “ejaculated” was replaced by “non-capacitated” throughout the whole MS

- Line 94-98: Very brief introduction is necessary for specie, number of males used, number of ejaculates and whether fresh, chilled or frozen-thawed and if ejaculates were always split

Response: This is now included.

- Line 100-101: please describe why some replicates were removed from analyses objectively if not included in Methods

Response: This is now elaborated in the M&M section.

- Line 152: remove semicolon

Response: It has been removed.

- Line 156-160: please restructure for flow

Response: This section has been restructured.

- Line 160: These 3 proteins seem to be important but results are not mentioned, likely due to restriction of journal scope etc. The reviewer suggests either elaborating on these novel findings or eliminating to submit in a separate manuscript.

Response: Indeed, we had to move the whole results paragraph describing the three proteins to supplementary data due to the Nature word count limit. This paragraph is now included in the results section.

- Line 173 (cite suppl. Data)

Response: Fig. S1 is now referenced.

- Line 175 (first mention of species in the manuscript unless reviewer missed it) needs to be made clear in intro

Response: This has been amended, please see our previous responses.

- Line 186-190: restructure run-on sentence for clarity and flow

Response: This is now restructured.

- Line 191: How would a block to polyspermy and sperm binding to oviductal epithelium lead the authors to conclude that Zn can reversibly alter sperm motility when both interactions primarily depend on plasma membrane function?

Response: Our current understanding of the Zinc release during fertilization is that Zn^{2+} binds to the molecules on the sperm surface (proteins and plasma membrane) and effectively changes the polarity of the sperm surface which results in the altered sperm binding affinity. We previously demonstrated that sperm binding to both the oviductal epithelium and zona pellucida is a zinc-dependent event, and these two affinities go against each other. Zinc signature 1 and 2 spermatozoa have a high affinity to the oviductal epithelium, more precisely to the glycans present on the surface, and a low affinity to ZP glycoproteins. Once zinc ions are released from the spermatozoa, as a capacitation-related change, the affinity for the oviductal epithelium decreases (and the newly hyperactivated spermatozoa are released) while ZP activity increases. This release is inhibited in the presence of high (2.0 mM) zinc ions. Finally, after zinc sparks, these two affinities are presumably switched again. As for the motility, it was shown that the higher flagellar Zn^{2+} concentration negatively correlated with sperm motility [10.1093/humrep/deu075, 10.1016/s0015-0282(99)00141-7], thus prohibiting spermatozoa from passing through ZP. Furthermore, zinc reverses sperm attraction to progesterone – quite literally chemorepelling their motility pattern towards the oocyte (doi: 10.1093/humrep/dex232).

- How do the authors suggest that a transcriptionally and translationally quiescent protamine compact cell can model neurodegenerative diseases? Testicular function and sperm quality may indirectly be related to transgenic models. Without further explanation, the implications may be beyond the scope of results.

Response: We are currently developing the spermatozoon as a non-traditional model to study Huntington's disease. Spermatozoa and neurons share many characteristics and features such as membrane properties. Many neuronal receptors are also present in spermatozoa, and spermatozoa share excitability function with neurons even though they lack some components of the synaptic mechanism. Sometimes, spermatozoa are referred to as "neurons with tails" (10.1017/s1464793103006407). Further, the brain and testis share the highest inter-tissue similarity in gene expression patterns, with the highest huntingtin (HTT) protein abundance of all body tissues. HTT plays a direct role in spermatogenesis. Unique, shared testicular pathology with reduced numbers of germ cells and abnormal seminiferous tubules was reported in both the HD patients and mutant HTT mouse models. In the transgenic HD minipig model, the progression of HD was accompanied by sperm deterioration which can be used for monitoring the progression of HD disease. This is just a shortlist of

implications of how a spermatozoon can be used for the study of neurodegenerative, in this can Huntington's disease and other somatic-reproductive comorbidities. We have just finished writing a review article titled "Spermatozoan Metabolism as a Non-traditional Model for the Study of Huntington's Disease", in which we discuss in detail the possibilities of the sperm model for study neurodegenerative disease. In addition to spermatozoa and testicular tissues, we are expanding this line of research with haploid male germ cell lines derived from induced pluripotent somatic cells carrying HTT mutation.

- Line 216: please provide a citation to support

Response: Citation is now provided

- Line 221: Do low abundant zincoproteins have less biological function? Not clear

Response: It was meant in a more general sense, i.e. that the protein abundance is an independent variable while the biological activity of a protein depends on its abundance, or as formulated in the MS, the biological activity of a zincoprotein will be a function of its abundance. Yes, we expect that the higher abundance of the protein will reflect in the higher biological activity performed by this protein, and vice versa. This is now clarified in the MS

- Line 256: How do we know these proteins are conserved across mammals? Study was not based on Zn homeostasis, semen handling or processing. Please help to complete the link between a germ cell and neurodegenerative disease model.

Response: We do not know that for certain unless more research is conducted, although many of the zincoproteins identified in our porcine studies can be found in the published mass spectrometric data sets generated with human sperm protein extracts. Regardless, the word "mammalian" was changed to "porcine". The importance of zinc and its homeostasis in reproduction in various species is now described in the introduction, and our findings add up to it further. Improvements in semen handling and processing are just some of the suggested follow-up studies. The link should be now more explicit, hopefully.

- Line 414: Please indicate which methods "Antibodies and Reagents" were purchased for. (reader does not know that WB) was being performed so difficult to read and follow

Response: This is now indicated.

- Line 451: Please briefly list capacitating conditions

Response: This is now listed.

- Line 453: What are incubation conditions for NC sperm?

Response: NC spermatozoa were directly processed after the last wash, i.e. either fixed for ICC and IBFC, or protein extraction was performed. We added a note about direct processing after the last washing step in M&M.

- Line 467: Secondary binding of what?

Response: This was meant as non-specific binding, i.e. binding of proteins elsewhere than to immobilized Zn²⁺. “Secondary” was replaced with “non-specific”

- Line 511: primary and secondary controls?

Response: This is now included.

- Line 520: was staining accomplished in suspension or mounted (methods)

Response: The staining was performed in a suspension, this is now mentioned in M&M.

- Line 521: Please include EX/EC parameters for each secondary, intensity and imaging instrumentation (LED epifluorescence etc), how many sperm analyzed and number of reps

Response: Epifluorescence microscopy was used solely for capturing purposes and to confirm labeling, with excitation using an LED white light source. Spermatozoa were analyzed using image-based flow cytometry as a more objective method. We used probes in blue (DAPI), green (AlexaFluor 488), and infrared (AlexaFluor 647) emission spectra. Excitation and emission parameters, intensity, and imaging instrumentation for image-based flow cytometry are provided in the M&M.

- Line 627: Justification for choosing probability 0.5?

Response: 0.5 is the default threshold value, which represents sufficient confidence for the prediction.

- Line 631: Move to methods

Response: Per Commun. Biol. guidelines, there has to be a dedicated section “Statistics and reproducibility” which is a part of M&M.

- Line 702: Please make x-axis more clear: e.g. make clear what 1-5, 6-10 etc means (# amino acids?)

Response: This is now amended

Supplemental Data

- FigS3: Suggest to label columns as non-capacitated and capacitated for ease

Response: The suggested change was implemented.

- Are these non-permeabilized sperm? (include in figure legend)

Response: These are all permeabilized spermatozoa. This is now included in the legend.

- Were appropriate primary and secondary controls used to identify specific vs. non-specific binding

Response: Yes, we always perform at least appropriate primary control, using non-immune species-specific sera. We have not observed any non-specific binding. There is, however, a caveat using rabbit polyclonal sera that from our experience, tend to bind non-specifically to the acrosomal content. To mitigate this, we can eliminate acrosomal signals from fluorometric signal quantification by using image masks, as described in the sections referring to image-based flow cytometry. We also performed secondary control as well to exclude the autofluorescence signal. This control is now included as fig S15.

- A: difficulty seeing red in the AR of Capacitated sperm (is it there)?

Response: We have included an inset with the CA2 channel only

- EF- don't see staining in PP of F

Response: Here are some additional images of F, the red channel only.

- FigS4: Why is there discrepancy in y-axis relative intensity among all 4 replicates? Can you please explain the bimodal distribution of reps 3 and 5

Response: We believe it has to do with unequal total sperm count among all four replicates as the result of post-sort gating. The apparent bimodal distribution is contributed by the “histogram smoothing” feature in our analysis software (see different levels of histogram smoothness of rep 3) in the combination with a lower # of sperm count in reps 3 and 5 when compared to reps 2 and 4.

- Fig S5 lacks sufficient description in legend to tie experimental objective to results depicted. Text is presented on P2 of supplemental material but visualization of band(s) of interest including loading

controls is difficult to interpret. Perhaps a square around area of interest to guide reader along with simple mean relative intensity graphic would be helpful.

Response: The text and the corresponding figure (now Fig 6) is now presented in the MS. The supplemental figure showing 8 replicates of WB as well as its legend was amended according to the reviewer's suggestions

- Fig S6: Please clarify that the 5 different CCIN populations are not discrete proteins (isoforms) but instead refer to localization patterns

Response: This is now clarified

- Fig S9: why the discrepancy in detection of clean bands between reps 1-4 and 5-8? How are the authors able to determine relative intensity using reps 5-8 when bands run together around 50-60kD, yet single bands appear in 1-4 around 60kD? Perhaps difficulty in solubility?

Response: We used ~2.5x higher load of zincoproteins in reps 5-8 resulting in more background in the zincoprotein fractions. For densitometric quantification, we used the intensity of the 60 kDa band consistently throughout all 8 replicates.

- Fig S10: A and B need to move above respected isolation techniques rather than on the sides of the figure. KCL seems to pull down much more protein in Capacitated but total protein is less, which indicates that either CCIN abundance is higher or more has gained additional access to proteins with non-specific binding. How do the authors image these bands for quantification given the likelihood of non-specificity around 60KD?

Response: As stated in the M&M, the KCl fraction was precipitated and the whole precipitate (an equivalent of ~100 mil spermatozoa) was loaded on the gel; as opposed to 20 mil sperm equivalent in TrX-100 and NaOH fractions. It is apparent from the CBB stained membrane that the total protein load is the highest in the KCl fraction. As for the imaging and quantification purposes, we collect chemiluminescence signal at different time intervals (see a different exposure time below). Regardless, we were only interested in the last, NaOH fraction of the sequential extraction sequence, as according to the reference we cite in the MS, CCIN should be most abundant in that particular fraction.

• Fig S11: Reps 2 and 4 seems somewhat consistent but 5 appears to be an outlier. Was rep 5 included in analyses?

Response: Yes, all five replicates were used in statistical analysis.

We would like to express our greatest appreciation to reviewer 1 for their very diligent, and in detail comments and suggestions that helped to increase the quality and comprehensibility of our study.

Reviewer #2 (Remarks to the Author):

Summary of manuscript:

The manuscript contains reliable data on analysis of Zinc interacting proteins. Proteins were isolated using IMAC with immobilized Zn²⁺. While Zn²⁺ chelation before IMAC resulted in no pull down of these proteins (a good way for showing that these proteins had Zn²⁺ dependent interaction).

One part of the manuscript deals with proteomic analysis of the Zinc protein interactome (1740 HUGO annotated proteins). These proteins were classified into molecular functionality or in which biological processes and pathways they are involved.

The second part deals with capacitation induced changes of the zinc protein interactome. Indeed 102 proteins showed significant changes. Three most observed functions or processes that showed changes were related to sperm motility, energy metabolism and signaling (all three known to change indeed during sperm capacitation).

Three significant changed zinc interacting proteins were phenotyped (an showed capacitation dependent changes in immune fluorescent signal intensity, or in immune fluorescent localization, or a change in protein abundance).

Of interest is also that Huntington's and Parkinson's disease pathways are involved in sperm processes studied. The authors hypothesize that sperm could be a relevant model for studying these diseases.

General points:

It is unusual to me to see a reference to previous work in the abstract. Is this allowed in the format of Comm. Biol.?

Response: You are right, the reference has been removed to comply with the journal guidelines

Sections: I am used to manuscripts with sections heading Introduction (for line 43-91) which is missing, Results line 93-162) which is also lacking. Is the format for Comm. Biol. different from this?

Response: It was in the format for Nature that the MS was originally submitted and then transferred to Commun. Biol. The section headings are now included in the current version which is also in line with Commun. Biol guidelines

The authors mentioned in line 250-252 that this study lacks to link the observation to functionality, but that their aim was to present a list of zinc interacting proteins and capacitation dependent changes thereof. The lack of functional data is indeed a shortcoming. I think that the lack of functional data is acceptable for this manuscript but in this section at least some ideas about how to study this shortcoming. For instance are the Zinc interacting protein changes dependent on protein kinase A activation or cholesterol efflux or on membrane potential (the latter can be tested on plasma membrane level or in the mitochondria). So I would request some lines in which the authors give the reader a hint in how to experimentally study the functionality of changes in zinc interacting proteins in sperm.

Response: We apologize for the confusion, L250-2 were referring to the three phenotyped proteins that we showed changed during the sperm capacitation, as a tool for validating our data. Indeed, further studies of the 102 zincoproteins significantly changing during the capacitation or at least for those of which function and localization have not been yet reported need to be further studied in order to get a clearer picture of how they are involved in it as well as to further the understanding of capacitation process. A sentence was added to clarify this.

Specific points:

page 3 ending in line 91 line 56-64 cholesterol efflux is a process that is slower (hour time scale) than the activation of SACY (10 minutes time scale). This part must be rephrased accordingly.

Response: Yes, indeed. This is now rephrased as per the reviewer's suggestion.

References in general: is not in consistent format regarding Capital letters or not sometimes all nouns are capitalized (for instance ref 1,2,3) and many times they are not (for instance 4, 5,6). Idem sometimes the nouns are spelled in full (for instance ref 3, 4) while sometimes they are abbreviated (for instance ref 1,2).

Ref 17, 27, 28: these citations contain the city and country of the journal *Reproduction*. I think that part should be skipped.

Response: We acknowledge this point risen by the reviewer. The references are in the Nature format as per journal guidelines. The authors believe that the final formatting (capitalization, etc will be done by production editors, alternatively by the authors during the proofreading, if the manuscript is accepted for publication).

REVIEWERS' COMMENTS:

Reviewer #1 (Remarks to the Author):

Comments to the Author

Summary

Authors of the manuscript have made substantial revisions that improve clarity and readability throughout. The Reviewer appreciates why initial submission included organization of material unique to previous journal submission prior to transfer, and thus reflected a substantial amount of reformatting. Barring the few comments below, there are no further major concerns.

Specific Comments

Title: The Reviewer understands that the mammalian sperm proteome is poorly characterized with databases lacking that require inference and approximations to draw conclusions regarding presumptuous functions in sperm. However, the Reviewer still encourages the authors to consider that although conserved functions support binding, motility and metabolic activities in sperm, these functions have not been explicitly challenged herein. The title would be more reflective of the scope of work to include some clarification such as the use of "consistent with" or "associated" so as not to unintentionally mislead readers into perceiving that these sperm functions were tested, especially with an N of 1.

Figure 3: Suggestion: the figure may become more clear if "Molecular Functions and Biological Processes" are indicated on the horizontal axis while "High and Low" are displayed on the vertical axis.

Line 152 response: it seems that this section contains some sentences that use commas and one long sentence that uses semicolons. May be best to have all commas for consistency.

Figure S5: The inset in A appears to be a mirror image (flipped). Also, please double check to make sure that the sperm in the inset are represented in the larger image. They may inadvertently be a different image.

Figure S9 response: although the Reviewer understands that 2.5X zincoproteins were loaded and thus increased band width, the 60kD bands in 5-8 do not appear discrete. The authors may want to clarify that this was a reference band and emphasize the use of CBB as was nicely illustrated in the rebuttal letter.

Reviewer #2 (Remarks to the Author):

The authors have addressed all points both from me and especially of reviewer 1.

I think this manuscript is now in an acceptable form to be published. May I add to this that a very recent study is published in *Front. Cell Dev. Biol.*, 18 February 2022 | <https://doi.org/10.3389/fcell.2022.836208>. It reports on the high enrichment of Huntington Disease proteins in protein complexes that form the of the perinuclear theca. This structure dissociates under disulfide bridge reducing conditions in a Zn²⁺ dependent way. This information may be useful and thus may referred to if the authors agree with me on this.

Response to review COMMSBIO-21-2810A

Once again, we thank the editorial board and both reviewers for thoughtful comments, which we followed closely as we revised our manuscript.

Reviewer #1 (Remarks to the Author):

Summary

Authors of the manuscript have made substantial revisions that improve clarity and readability throughout. The Reviewer appreciates why initial submission included organization of material unique to previous journal submission prior to transfer, and thus reflected a substantial amount of reformatting. Barring the few comments below, there are no further major concerns.

Response: We are pleased with Reviewer 1 feedback, and thankful for their appreciation.

Specific Comments

Title: The Reviewer understands that the mammalian sperm proteome is poorly characterized with databases lacking that require inference and approximations to draw conclusions regarding presumptuous functions in sperm. However, the Reviewer still encourages the authors to consider that although conserved functions support binding, motility and metabolic activities in sperm, these functions have not been explicitly challenged herein. The title would be more reflective of the scope of work to include some clarification such as the use of “consistent with” or “associated” so as not to unintentionally mislead readers into perceiving that these sperm functions were tested, especially with an N of 1.

Response: The title was changed according to the Reviewer’s suggestion

Figure 3: Suggestion: the figure may become more clear if “Molecular Functions and Biological Processes” are indicated on the horizontal axis while “High and Low” are displayed on the vertical axis.

Response: Vertical axis was added as per the Reviewer’s suggestion to increase the clarity of Figure 4 (former Figure 3)

Line 152 response: it seems that this section contains some sentences that use commas and one long sentence that uses semicolons. May be best to have all commas for consistency.

Response: It is a common practice to have commas and semicolons in compound sentences. Semicolons separate three or more items in a series that already have commas in them so that the sentence is not confusing to a reader. To our best knowledge, the punctuation in this compound sentence is according to English syntax rules.

Figure S5: The inset in A appears to be a mirror image (flipped). Also, please double check to make sure that the sperm in the inset are represented in the larger image. They may inadvertently be a different image.

Response: This is now fixed.

Figure S9 response: although the Reviewer understands that 2.5X zincoproteins were loaded and thus increased band width, the 60kD bands in 5-8 do not appear discrete. The authors may want to clarify that this was a reference band and emphasize the use of CBB as was nicely illustrated in the rebuttal letter.

Response: This is now clarified in Figure S11 (former Fig S9) as per the Reviewer's suggestion. The same clarification was done for Fig S12, so it is clear that the higher signal intensity was caused by the higher total protein load rather than CCIN enrichment in that particular fraction.

Reviewer #2 (Remarks to the Author):

The authors have addressed all points both from me and especially of reviewer 1.

Response: We are very happy to hear that.

I think this manuscript is now in an acceptable form to be published. May I add to this that a very recent study is published in Front. Cell Dev. Biol., 18 February 2022 | <https://doi.org/10.3389/fcell.2022.836208>. It reports on the high enrichment of Huntington Disease proteins in protein complexes that form the of the perinuclear theca. This structure dissociates under disulfide bridge reducing conditions in a Zn²⁺ dependent way. This information may be useful and thus may referred to if the authors agree with me on this.

Response: We thank Reviewer 2 for pointing this new study out. It has caught our attention as well and this paper was a subject of the discussion at one of our recent lab meetings. Yes, we believe that the study is very relevant to current MS and is now included in it.